

# Evaluating biases in filter-based aerosol absorption measurements using photoacoustic spectroscopy

Nicholas W. Davies[1,2], Cathryn Fox[2], Kate Szpek[2], Michael I. Cotterell[1,2], Jonathan W. Taylor[3], James D. Allan[3,4], Paul I. Williams[3,4], Jamie Trembath[5], Jim M. Haywood[1,6] and Justin M. Langridge[2]

[1]College of Engineering, Mathematics and Physical Sciences, University of Exeter, Exeter, EX4 4QF, United Kingdom
[2]Observation Based Research, Met Office, Exeter, EX1 3PB, United Kingdom
[3]Centre for Atmospheric Science, School of Earth and Environmental Sciences, University of Manchester, Manchester, M13 9PL, United Kingdom
[4]National Centre for Atmospheric Science, University of Manchester, Manchester, M13 9PL, United Kingdom
[5]Facility for Airborne Atmospheric Measurements, Cranfield, MK43 0AL, United Kingdom
[6]Earth System and Mitigation Science, Met Office Hadley Centre, Exeter, EX1 3PB, United Kingdom

*Correspondence to*: Justin M. Langridge (justin.langridge@metoffice.gov.uk)

**Abstract.** Biases in absorption coefficients measured using a filter-based absorption photometer (Tricolor Absorption Photometer, or TAP) at wavelengths of 467, 528 and 652 nm are evaluated by comparing to measurements made using photoacoustic spectroscopy (PAS). We report comparisons for ambient sampling covering a range of aerosol types including urban, fresh biomass burning and aged biomass burning. Data are also used to evaluate the performance of three different TAP correction schemes. We found that photoacoustic and filter-based measurements were well correlated, but filter-based measurements generally overestimated absorption by up to 45 %. Biases varied with wavelength and depended on the correction scheme applied. Optimal agreement to PAS data was achieved by processing the filter-based measurements using the recently developed correction scheme of Müller et al. (2014), which consistently reduced biases to 0–17 % at all wavelengths. The biases were found to be a function of the ratio of organic aerosol mass to light-absorbing carbon mass although applying the Müller et al. (2014) correction scheme to filter-based absorption measurements reduced the biases and the strength of this correlation significantly. Filter-based absorption measurement biases led to aerosol single-scattering albedos that were biased low by up to 0.07 and absorption Ångström exponents (AAE) that were in error by ± 0.54. The discrepancy between the filter-based and PAS absorption measurements is lower than reported in some earlier studies, and points to a strong dependence of filter-based measurement accuracy on aerosol source type.

## 1 Introduction

Aerosol-radiation interactions are estimated to contribute a global mean effective radiative forcing of −0.45 (−0.95 to +0.05) W m$^{-2}$, offsetting a potentially significant but poorly constrained fraction of the positive effective radiative forcing associated with greenhouse gases (2.26 to 3.40) W m$^{-2}$ (Myhre et al., 2013a). One of the major factors governing the uncertainty in estimates of aerosol direct radiative forcing is the poorly constrained aerosol single scattering albedo (SSA),



defined as the ratio of aerosol scattering to total extinction (Loeb and Su, 2010; McComiskey et al., 2008; Sherman and McComiskey, 2018). Accurate determination of aerosol SSA is limited by uncertainties in aerosol absorption estimates, which could potentially be underestimated by up to a factor of two in global climate models (Shindell et al., 2013; Stier et al., 2007).

The main types of absorbing aerosol include black carbon (BC) and light-absorbing organic carbon, commonly referred to as brown carbon (BrC) (e.g. Myhre et al., 2013a). On a global scale, aerosol absorption is dominated by BC, a carbonaceous product formed during incomplete combustion, which may exert the next largest positive radiative forcing after carbon dioxide (Stocker et al., 2013). BC absorbs strongly across visible wavelengths and contributes an estimated 0.71 (0.09 to

1.26) W m$^{-2}$ to aerosol direct radiative forcing (Bond et al., 2013). In recent years, BrC has received increasing attention as a climate forcing agent (e.g. Feng et al., 2013). Sources of BrC include primary emissions during biomass and biofuel combustion as well as secondary production via photo-oxidation of volatile organic compounds (Andreae and Gelencsér, 2006; Wang et al., 2018). BrC has been found to absorb strongly towards ultraviolet wavelengths, although the strength and wavelength dependence of this absorption is uncertain, due in part to the wide range of compounds that this term

encompasses, many of which are poorly characterised (Andreae and Gelencsér, 2006; Lack et al., 2012b; Pokhrel et al., 2017). Climate models generally only crudely represent the optical properties of BC and BrC and their evolution with time. For example, while the Met Office Hadley Centre HadGEM3 model treats the internal mixing of aerosol components, the real and imaginary parts of the refractive index of organic carbon that are used to calculate the radiative properties of the composite aerosol are fixed (e.g. Johnson et al., 2016). In order to address this deficiency, stronger observational constraints

are first required (e.g. Alexander et al., 2008; Bond et al., 2013; Liu et al., 2014; Myhre et al., 2013b; Saleh et al., 2014; Wang et al., 2018).

Over the course of several decades, filter-based absorption photometry has been used to measure aerosol absorption coefficients. The approach has considerable benefits including that it is relatively inexpensive, portable and capable of

unattended measurements for long periods of time (Baumgardner et al., 2012). Filter-based instruments measure the light transmittance across a filter continuously, which changes as particles are deposited onto the filter, providing a measure of aerosol absorption (see Sect. 2.1) (Lin et al., 1973). Absorption coefficients determined using filter-based absorption photometry can be subject to measurement artefacts due to (i) scattering of light away from the light-detector leading to erroneous apparent absorption (Bond et al., 1999) and (ii) enhanced absorption as particles are deposited onto the filter. The

latter leads to multiple scattering between the particles and the filter medium, providing multiple opportunities for absorption. The enhancement is complex to characterise and depends on the filter loading such that an increase in the number of deposited absorbing particles reduces the multiple scattering between the filter and particle layers (Bond et al., 1999; Liousse et al., 1993; Weingartner et al., 2003) leading to lower absorption coefficients for highly loaded filters (Weingartner et al., 2003). A number of empirical and semi-empirical correction schemes have been derived to correct for





the aforementioned artefacts. This study will focus on correction schemes derived for use with glass-fibre Pallflex E70-2075W filters that have been used widely with the Particle Soot Absorption Photometer (PSAP, Radiance Research) (Bond et al., 1999; Müller et al., 2014; Virkkula, 2010; Virkkula et al., 2005). These correction schemes are also valid for similar instruments using this filter substrate, for example the Tricolor Absorption Photometer (TAP, Brechtel Manufacturing) used
in this study and described in Sect. 2.3.2 (Ogren et al., 2017).

Another potentially significant measurement artefact is due to liquid-like organic aerosols spreading across the filter fibres (Lack et al., 2008). The mechanisms proposed for this artefact include a change in the physical shape and therefore optical properties of deposited particles, or a coating effect whereby deposited particle absorption is enhanced via a lensing effect
(Cappa et al., 2008; Lack et al., 2008; Subramanian et al., 2007). Although recognised as potentially significant, there are no empirical corrections to account for these artefacts.

Previous work has examined the magnitude of biases in filter-based absorption measurements. For example, Lack et al. (2008) found PSAP absorption coefficients were biased high in the range 12 % to over 200 % at 532 nm compared to
photoacoustic spectroscopy measurements for aerosols over the Gulf of Mexico, which included BC, nitrate, sulphate and organic aerosols from shipping emissions. The PSAP biases were found to be positively correlated to the organic aerosol mass concentration and even more strongly correlated to the ratio of the organic aerosol to light-absorbing carbon mass. To verify these measurements, Cappa et al. (2008) performed laboratory experiments using secondary organic aerosol (SOA) derived from the ozonolysis of α-pinene, which had a SSA > 0.998 at 532 nm. A key finding of this study was that for
external mixtures of SOA and soot, the PSAP absorption could be biased high by a factor 2.6, consistent with the findings of Lack et al. (2008). Cappa et al. (2008) also found that the magnitude of the absorption bias was strongly dependent upon the filter transmittance and that the bias was both immediate (clean filter) and cumulative (filter previously exposed to absorbing material). The results from both of these studies (Cappa et al., 2008; Lack et al., 2008) were independent of the correction scheme applied (Bond et al., 1999; Virkkula et al., 2005).

More recently, Subramanian et al. (2010) derived the BC mass absorption coefficient (MAC) at 660 nm for fresh and 1–2 day-old aerosol emissions in and around Mexico City by dividing the absorption coefficients measured using a PSAP by the BC mass concentrations measured using a single particle soot photometer (SP2, Droplet Measurement Technologies). For the fresh emissions, they found a ~50 % enhancement in their measured BC MAC relative to the value reported by Bond and
Bergstrom (2006), whose review was based on an extensive range of measurements. The BC MAC bias was attributed in part to an over-estimation of the absorption coefficients measured by the PSAP due to externally mixed liquid-like organic matter. However, the BC MAC values for the relatively thickly coated, aged BC further from the city were in line with those estimated by Bond and Bergstrom (2006), which the authors suggested may indicate that biases in filter-based measurements





relating to high organic aerosol loading may only be present when organic aerosol is externally mixed with BC (Subramanian et al., 2010).

Using a similar methodology, McMeeking et al. (2011) derived the BC MAC at 550 nm using PSAP and SP2 measurements
for urban pollution aerosols around the UK, reporting organic aerosol mass concentrations in the range 1–7 µg m$^{-3}$. The work by Lack et al. (2008) indicates that a positive absorption bias of up to 50 % would be expected at these loadings, however no bias in the BC MAC was observed. McMeeking et al. (2011) postulated that this result could be due to limitations in the PSAP and SP2 measurements or a physical effect whereby absorption enhancements due to coatings were offset by the collapse of fractal BC aggregates (McMeeking et al., 2011). Indeed, another explanation for this discrepancy
could have been that the organic aerosol sampled here was not quasi-liquid like and contributed different biases to those seen in previous studies.

Biases in filter-based absorption photometry measurements can limit the accurate determination of key climate-relevant parameters including, for example, the aerosol SSA and AAE (e.g. Sherman and McComiskey, 2018). Mason et al. (2018)
compared PAS to filter-based absorption measurements of wildfires and agricultural fires over the continental U.S.A. during August and September 2013, which included a PSAP and a Continuous Light Absorption Photometer (CLAP) (Ogren et al., 2017). All PSAP and CLAP data were corrected using the Bond et al. (1999) and Ogren (2010) corrections. Biases in filter-based measurements were evaluated by comparison to PAS measurements, which were in the range 0.61 to 1.24, dependent on measurement wavelength (405, 532 and 660 nm). Mean SSA and AAE values derived using filter-based absorption
photometry were found to be in error by up to 0.03 and 0.7, respectively, compared to PAS.

Further, Backman et al. (2014) assessed the sensitivity of the PSAP-derived AAE to the Bond et al. (1999) and Virkkula (2010) correction schemes for measurements recorded on the central Highveld in South Africa, where emissions were dominated by fossil-fuel burning activities including from coal-fired power plants. They found that the AAE varied between
1.34 to 1.96 dependent upon the PSAP correction scheme applied, which led to different conclusions regarding the aerosol composition and source (Backman et al., 2014).

Despite this body of previous work, there remains significant uncertainty related to the magnitude of biases in filter-based absorption measurements, particularly regarding dependence on source type and the correction scheme applied. The aim of
this study is to address this gap. We assess biases by comparing absorption coefficients determined using multi-wavelength TAP and photoacoustic instruments during a series of research flights aboard the UK Facility for Airborne Atmospheric Measurements (FAAM) BAe-146 aircraft. Aerosol sources sampled include urban aerosol emissions over London, fresh biomass burning aerosol (BBA) over West Africa and aged BBA over the Southeast Atlantic Ocean. We follow the methodology of Lack et al. (2008) by looking at the absorption biases as a function of organic aerosol concentration,





extending their study by looking at a greater range of wavelengths and aerosol sources as well as evaluating additional correction schemes, namely those developed by Virkkula (2010) and Müller et al. (2014). We then assess the impact that biases in filter-based absorption photometry have on the aerosol SSA and AAE. To our knowledge, this is the first study to simultaneously evaluate the Bond et al. (1999), Virkkula (2010) and Müller et al. (2014) correction schemes for ambient

aerosol sampling across multiple aerosol types.

## 2. Methodology and measurements

### 2.1 Principles of filter-based absorption photometry

Filter-based absorption photometers measure the light transmitted through a filter as particles are deposited onto the filter such that the attenuation can be defined as

$I = -ln\left(\frac{I_s}{I_r}\right),$      (1)

where $I_s$ and $I_r$ are the intensities of light transmitted through a filter corresponding to a sample spot (i.e. an area of the filter with deposited aerosols) and reference spot (i.e. an area of the filter without deposited aerosols), respectively (Ogren et al., 2017). The absorption coefficient can thus be determined using

$b_{ap}^{raw} = \frac{A}{Q\Delta t}(I(t + \Delta t) - I(t)),$      (2)

where $A$ is the area of the aerosol deposited onto a filter, $Q$ is the flow rate of the aerosol-laden stream pulled through a filter, $\Delta t$ is the time between successive measurements of light attenuation and $I(t)$ and $I(t + \Delta t)$ are the light attenuations at times $t$ and $t + \Delta t$ (Ogren et al., 2017). To correct $b_{ap}^{raw}$ for apparent and enhanced absorption, we applied the correction schemes developed by Bond et al. (1999), Virkkula, (2010) and Müller et al. (2014), which will be referred to as $b_{ap}^{B1999}$, $b_{ap}^{V2010}$ and $b_{ap}^{M2014}$ respectively. See Sect. 2.1.1–2.1.3.

### 2.1.1 The Bond et al. (1999) correction scheme (B1999)

The Bond et al. (1999) correction scheme was developed empirically by comparing PSAP absorption coefficients to reference absorption coefficients determined using the difference between extinction as measured by an optical extinction cell and scattering coefficients measured using a nephelometer. Calibration aerosols included polydisperse nigrosin and ammonium sulphate. This correction scheme was updated by Ogren (2010). Bond et al. (1999) found that

$b_{ap}^{B1999} = f(Tr)b_{ap}^{raw} - sb_{sp},$      (3)

with

$f(Tr) = \frac{0.85}{K_2(1.0796Tr+0.71)},$      (4)

$s = \frac{K_1}{K_2},$      (5)



and where $b_{sp}$ is the scattering coefficient, $K_1$=0.02, $K_2$=1.22 and $Tr$ is the normalised filter transmittance, defined as (Ogren et al., 2017)

$$Tr = \frac{I_s(t)/I_r(t)}{I_s(0)/I_r(0)}. \tag{6}$$

This correction scheme was derived at the wavelength 550 nm and is generally assumed to apply over the entire range of
visible wavelengths, though there is no empirical basis for this (Bond et al., 1999; Ogren, 2010).

### 2.1.2 The Virkkula (2010) correction scheme (V2010)

The Virkkula et al. (2005) correction scheme and the subsequent Virkkula (2010) erratum were derived for the PSAP wavelengths 467, 530 and 660 nm, which is reflected by the $f(Tr, \lambda)$ term described below. The scheme was derived by comparing absorption coefficients determined using a multi-wavelength PSAP to those measured using either photoacoustic
spectroscopy or to absorption derived by subtracting scattering from extinction measurements (Virkkula et al., 2005). Calibration aerosols included kerosene soot, graphite, diesel soot, ammonium sulphate and polystyrene latex spheres. Virkkula (2010) proposed that

$$b_{ap}^{V2010} = f(Tr, \lambda)b_{ap}^{raw} - sb_{sp}, \tag{7}$$

where
$$f(Tr, \lambda) = k_0 + k_1(h_0 + h_1\omega_0)ln(Tr), \tag{8}$$

and where $k_0$, $k_1$, $h_0$, $h_1$ and $s$ are wavelength dependent constants and $\omega_0$ is the wavelength dependent SSA. The values of the constants used in this study were taken directly from Table 1 in Virkkula (2010), which are provided in Table 1. The wavelengths at which these constants were derived differ to those used in the TAP by 2 nm and 8 nm at the green and red wavelengths, respectively. It is unclear how these constants depend on wavelength. To assess the impact that this wavelength
mismatch might have on the absorption coefficients derived using the V2010 correction scheme, the single-wavelength V2010 constants were also applied to TAP measurements. These were taken from Table 1 in Virkkula (2010) and are provided in the fifth column of Table 1. This was found to have a minor impact on the results of this study as discussed in Sect. 3. The Virkkula (2010) correction is an iterative correction scheme due to its dependence on the SSA. Hence the algorithm was run 10 times for each time-step, which was sufficient for the absorption coefficient to converge to a single
value with a precision better than 0.001 Mm$^{-1}$.

### 2.1.3 The Müller et al. (2014) correction scheme (M2014)

The constrained two-stream (CTS) algorithm developed by Müller et al. (2014) includes a two-stream radiative transfer model that explicitly accounts for the optical properties of the filter substrate and deposited particles and is constrained by either the Bond et al. (1999), Virkkula et al. (2005) or Virkkula (2010) parameterisations. This section covers the key
equations from Müller et al. (2014) to show how they have been implemented in this study and the reader is referred to





Müller et al. (2014) for a full derivation. The M2014 correction scheme makes use of the relationship between the absorption coefficient and the change in particle absorption optical depth, $\delta_{ap}$, on the filter medium between two measurements separated by a time-step $\Delta t$, as represented by:

$$b_{ap}^{M2014} = \frac{A}{Q\Delta t}\left(\delta_{ap}(t + \Delta t) - \delta_{ap}(t)\right), \tag{9}$$

For each time-step, $\delta_{ap}$ was calculated iteratively by minimising the difference between the measured total optical depth, $\delta_{tot}$ (filter + particles) and the relative optical depth, $\delta_{CTS}$, which is the change in total optical depth of the filter system after collecting a particle relative to the unloaded filter. A Newton-type solver was applied, as suggested by Müller et al. (2014), and required ten iterations to converge to a precision better than 0.001 Mm$^{-1}$. $\delta_{tot}$ was calculated from measurements of the filter, with and without aerosol, using Eq. 1. The equations outlined in Müller et al. (2014) were used to calculate $\delta_{CTS}$ and

are included here for clarity.

$$\delta_{CTS} = \frac{F_s^{exp}\delta_{sp} + F_a^{exp}\delta_{ap}}{F_f^{mod}}, \tag{10}$$

where $\delta_{sp}$ is the particle scattering optical depth, calculated using

$$\delta_{sp} = \frac{Q\Delta t}{A}\sum_{t=0}^{t} b_{sp}(t), \tag{11}$$

$$F_s^{exp} = a_5 + (a_0 + a_1 g_p)e^{-\left(\frac{ln(\delta_{sp}) + a_4^2}{a_3 + a_4 g_p}\right)^4}, \tag{12}$$

where $a_0 = 0.1509$, $a_1 = -0.1611$, $a_2 = 4.5414$, $a_3 = -5.7062$, $a_4 = -1.9031$, $a_5 = 0.01$ and $g_p$ is the average weighted particle asymmetry parameter (see Eq. 23). Using the V2010 empirical correction,

$$F_{a,V2010}^{exp} = \frac{1}{\delta_{ap}}\sqrt{\left(\frac{c_1}{c_2 h_0}\right)^2 - \frac{2\delta_{ap}}{c_2 h_0}} + \frac{c_1}{c_2 h_0}, \tag{13}$$

where $c_1$, $c_2$, $h_0$, $h_1$ and $s$ correspond to the wavelength dependent constants $k_0$, $k_1$, $h_0$, $h_1$ and $s$ as defined in Sect. 2.1.2, corresponding to the Virkkula (2010) parameterisation. Finally,

$$F_f^{mod}\left(\delta_{ap}, \delta_{sp}, g_p\right) = \frac{\delta(\delta_{ap}=0, \delta_{sp}, g_p) + \delta(\delta_{ap}, \delta_{sp}=0, g_p)}{\delta(\delta_{ap}, \delta_{sp}, g_p)}, \tag{14}$$

where

$$\delta\left(\delta_{ap}, \delta_{sp}, g_p\right) = -ln\left(T_{2L}\left(\delta_{ap}, \delta_{sp}, g_p\right)\right) + ln\left(T_{2L}\left(\delta_{ap} = 0, \delta_{sp} = 0, g_p = 0\right)\right), \tag{15}$$

$$T_{2L} = \frac{T_1 T_2}{1 - R_1(1 - T_2)}, \tag{16}$$

and $T_1$ and $T_2$ represent the filter transmittances of the particle-loaded and particle-free layers, respectively. These are

represented by layers 1a and 1b in Müller et al. (2014), respectively. The filter transmittance and reflectance are given by

$$T = \frac{2}{[2 - \omega_0(1 + g)]sinh(K\delta_e/\mu_1)/K + 2cosh(K\delta_e/\mu_1)} \tag{17}$$

and



$$R = \frac{\omega_0(1-g)\sinh\left(K\delta e/\mu_1\right)/K}{[2-\omega_0(1+g)]\sinh\left(K\delta e/\mu_1\right)/K + 2\cosh\left(K\delta e/\mu_1\right)},$$  (18)

where

$$\delta_e = \chi\delta_{sf} + \delta_{sp} + \chi\delta_{af} + \delta_{ap},$$  (19)

$$K = \sqrt{(1-\omega_0)(1-\omega_0 g)},$$  (20)

$$\omega_0 = \frac{\chi\delta_{sf} + \delta_{sp}}{\chi\delta_{sf} + \delta_{sp} + \chi\delta_{af} + \delta_{ap}},$$  (21)

and

$$g = \frac{\chi g_f \delta_{sf} + g_p \delta_{sp}}{\chi\delta_{sf} + \delta_{sp}}.$$  (22)

The filter scattering optical depths used in this study were $\delta_{sf}^{467} = 7.76$, $\delta_{sf}^{530} = 7.69$ and $\delta_{sf}^{660} = 7.34$ and the filter absorption optical depths used were $\delta_{af}^{467} = 0.033$, $\delta_{af}^{530} = 0.038$ and $\delta_{af}^{660} = 0.019$, as measured by Müller et al. (2014) for the same type of filters. Small differences between wavelengths that the filter optical properties were measured at by Müller et al. (2014) (467, 530, 660 nm) compared to those at which the TAP measures (467, 528, 652 nm) were assumed to be negligible. Following the nomenclature of M2014, for filter layer 1 (the particle-loaded filter layer) $\chi = 0.2$ and for layer 2 (the unloaded filter layer) $\chi = 0.8$. This assumes that the particle penetration depth into the filter was 20 % and accounts for the fractional filter optical depths corresponding to each layer. The value used for $\mu_1$ was $1/\sqrt{3}$. The value $g_p$ is the average weighted asymmetry parameter of all particles deposited onto the filter, given by

$$g_p = \frac{\sum_i b_{sp}^i g_p^i}{\sum_i b_{sp}^i}$$  (23)

where $i$ represents the $i$ th ensemble of particles with scattering coefficient $b_{sp}^i$. This is different to the equation presented by Müller et al. (2014) who recommended using individual particle scattering cross sections (as opposed to ensemble scattering coefficients). We used Eq. 23 as opposed to the recommended formulation because nephelometer measurements represent an ensemble. In this study, bulk asymmetry parameters (i.e. corresponding to an ensemble of particles) were calculated for each time-step using the parameterisation

$$g_p = -7.143889 b_{sp}^3 + 7.464439 b_{sp}^2 - 3.96356 b_{sp} + 0.9893,$$  (24)

where $b_{sp}$ is the backscattering ratio measured using a nephelometer (Andrews et al., 2006; Müller et al., 2014).

To confirm the accuracy of the implementation of the M2014 algorithm used during this analysis, equations 15–22 were used to reproduce the results in Fig. 6 of the Müller et al. (2014) study, which were verified against intermediate results (T. Müller, personal communication, 2016).





### 2.3 Measurements and instrumentation

All measurements presented in this study were made aboard the UK's BAe-146-301 large Atmospheric Research Aircraft (ARA) operated by the Facility for Airborne Atmospheric Measurements (FAAM; www.faam.ac.uk). The aircraft is capable of carrying 3 crew, 18 scientists and a total scientific payload of up to 4000 kg with a range up to 3700 km. This section provides information on the filter-based, photoacoustic, nephelometer and aerosol composition instrumentation used aboard the aircraft and introduces the environments in which measurements were made.

#### 2.3.1 Aerosol sampling and conditioning

An important strength of this dataset is that the TAP, PAS and cavity ring-down spectrometer (CRDS) instruments used to sample aerosol optical properties all shared a common sample inlet and were subject to the same flow conditioning. Aerosols were drawn into the aircraft through a modified Rosemount inlet (Trembath et al., 2012). The aerosol-laden stream was first dried to < 20 % relative humidity (Permapure, PD100T-12MSS) and then passed through a scrubber (MAST Carbon) to remove absorbing gaseous impurities such as ozone and nitrogen dioxide. An impactor removed particles with aerodynamic diameter > 1.3 μm (Brechtel, custom design). A series of flow splits (Brechtel 1110 and 1104) evenly distributed the aerosol-laden stream between the suite of instruments, which each sampled the aerosol at a flow rate of 1 L min$^{-1}$, as shown in Fig. 1. All measurements were corrected to standard temperature and pressure (PAS, CRDS and TAP: 20 °C and 1013 mb).

#### 2.3.2 Tricolor Absorption Photometer

The TAP is a commercially available (Brechtel) version of the Continuous Light Absorption Photometer (CLAP), described by Ogren et al. (2017). The TAP comprises eight sample filter spots and two reference filter spots. The aerosol-laden air passes through one sample spot at a time, which allows for eight times the filter lifetime compared to single-spot photometers. The filtered air is re-circulated through one of the reference spots to enable the attenuation calculation (see Eq. 1) (Ogren et al., 2017). Upon reaching a pre-defined filter transmittance set point, the TAP automatically changes to the next available sample filter spot. We used 47 mm diameter Pallflex (E70-2075W) glass-fibre filters, which were nominally identical to the filters used to derive the correction schemes applied in this study (see Sect. 2.1.1–2.1.3). The TAP provides measurements at three wavelengths with peaks centred at 467, 528 and 652 nm, which allows the spectral dependence of climate relevant parameters such as the SSA and AAE to be evaluated (Sect. 3.3). The LEDs are cycled through each wavelength once per second, providing absorption measurements at 1 Hz at all wavelengths. The inlet of the TAP is heated to 35.2 ± 0.2 °C to minimise the effects of changing temperature and to prevent water condensing onto the filter. The built-in digital low-pass filter was disabled in all of our measurements to enable calculation of the absorption coefficients from the raw photodiode measurements, as it was unclear how the low-pass filter impacted the measurements. To understand the impact of this on instrument sensitivity, the TAP was run for ~3 h while it sampled filtered room air to characterise the noise in the system. Uncorrected absorption coefficients, $b_{ap}^{raw}$, were calculated at 1 Hz, and the average and standard deviation for





each time interval $\Delta t$ ($1 < \Delta t < 1000$ s) were calculated. The 1-sigma detection limits at 30 s averaging time were 0.71, 1.37 and 0.89 Mm$^{-1}$ at wavelengths 658, 528 and 467 nm, respectively. Ogren et al. (2017) calculated the mean 1-sigma detection limit for their 28 instruments over all three wavelengths to be 0.33 Mm$^{-1}$. The difference between the detection limits measured in this study and that presented in Ogren et al. (2017) is likely due to running without low-pass digital filtering in

the current study. TAP internal particle losses were estimated to be < 1 % for particles with diameters in the range 0.03–2.5 μm (Ogren et al., 2017).

To determine the areas of the spots resulting from particle deposition onto the filter, nigrosin (product number 198285-100G) was atomised from solution, dried to < 10 % relative humidity using a silica gel diffusion drier (Topas, DDU-570) and

sampled by the TAP. The areas of the eight sample spots were determined by measuring the number of pixels corresponding to the diameters in a magnified digital photograph, which yielded areas in the range 32.4–36.8 mm$^2$. Filter spot sizes were determined using nigrosin rather than from the ambient aerosol samples themselves as the spot edges were more clearly defined. This analysis used the areas determined using the clearly defined nigrosin spots, and therefore provides a lower limit of area and thus absorption coefficient (see Eq. 2).

**2.3.3 Photoacoustic and cavity ring-down spectrometers**

The photoacoustic and cavity ring-down spectrometers used in this study were based on the designs by Lack et al. (2012) and Langridge et al. (2011), respectively and are described in detail in Davies et al. (2018) and Szpek et al. (in preparation). PAS measures absorption directly for aerosols in their suspended state (Arnott et al., 1999). The PAS principle relies on converting energy from a light source into sound. Light-absorbing media, such as aerosol, transfer electromagnetic energy

into thermal energy that heats the surrounding air. This gaseous heating generates a pressure wave, which is detected by a microphone located within the PAS cell. The amplitude of the microphone signal is related to the sample absorption coefficient through calibration (Arnott et al., 1999; Davies et al., 2018; Moosmüller et al., 2009).

Much of this analysis relies on accurate PAS absorption measurements and thus we focus here on describing the uncertainty associated with these measurements. The total PAS measurement uncertainty is comprised of the measurement precision and

accuracy. The PAS measurement precision was derived by evaluating the minimum sensitivities of the suite of PAS instruments in a similar way to the TAP, as described in Sect. 2.3.2, and were in the range 0.01–0.06 Mm$^{-1}$ for 30 s averaging across the range of cells used. The minimum sensitivities of the suite of CRDS cells were evaluated in the same way and were found to be 0.02–0.05 Mm$^{-1}$ across the range of cells used.

The accuracy of PAS absorption measurements was determined by three factors: (i) uncertainty in the ozone calibration, (ii) uncertainty in corrections applied to account for the PAS microphone pressure sensitivity and (iii) uncertainty in subtraction of background noise which arose primarily from laser heating of the PAS cell optical windows. We consider each of these in turn below.





The accuracy of the PAS ozone calibration has previously been evaluated in laboratory experiments that compared measured and modelled absorption and extinction cross sections of strongly-absorbing nigrosin aerosol. This analysis showed the PAS calibration accuracy to be better than 8 % and the accuracy of the CRDS instruments used in this study to be better than 2 %

(Davies et al., 2018).

The second source of PAS measurement uncertainty was due to the PAS microphone sensitivity to pressure, which was evaluated by performing ozone calibrations at several pressures in the range 600–1000 mb (typical of those encountered during airborne operation). The measured PAS microphone sensitivities were fit to a linear trend across this range and

normalised to yield a correction factor that varied from 0.83 (600 mb) to 1.00 (1000 mb). The uncertainty introduced by applying this pressure-dependent correction to PAS calibrations was estimated by propagating the 1σ fitting uncertainties in the linear regression between the calibration factors to in-flight PAS measurements, which led to uncertainties in PAS absorption coefficient measurements of 0.0–1.2 %. The smallest uncertainties were associated with measurements around 1000 mb where there was no correction applied and largest for relatively low pressures where the largest correction was

applied.

The third source of PAS measurement uncertainty was due to subtraction of window-generated background noise, which is unstable for airborne operation due to its dependence on pressure. To account for this, in-flight background noise is typically characterised by periodically measuring a filtered-air stream for 30 s every 300 s. These measurements are then used post-

flight to derive a background correction as a function of pressure. To evaluate the uncertainty introduced by this background noise correction, we took continuous PAS measurements of filtered-air in the laboratory and varied the pressure within the PAS cells over the range encountered during airborne operation. This laboratory PAS dataset was then processed to mimic in-flight conditions, with 30 s windows of data every 300 s being used to derive a continuous pressure-dependent background correction. Examining the difference between the continuous filtered-air measurements and the synthetically

generated background data series provided the uncertainty in the background noise correction under variable pressure conditions. The uncertainty in the background noise correction was found to be normally distributed, with a 1σ width of 1.27 %. This uncertainty was propagated through in-flight PAS data processing to derive the uncertainties introduced to airborne PAS absorption coefficient measurements from the background noise subtraction. The uncertainty depended on the strength of aerosol absorption and was found to be 0.2, 2.0 and 20.5 % at 100, 10 and 1 Mm$^{-1}$ respectively.

The total uncertainty in PAS measurements is the combination of the measurement precision and accuracy, including the PAS calibration accuracy, the pressure-dependent calibration correction uncertainty and the background noise correction uncertainty. These factors were combined in quadrature, leading to total PAS measurement uncertainties of 23.1 % for 1 Mm$^{-1}$ absorption coefficient measurements (independent of pressure) and approximately 8.0 % for 100 Mm$^{-1}$. These


uncertainties are in-line with previous estimates for airborne PAS measurements, which were found to be ± 5 % for ground-based measurements with an additional ± 0.5 Mm$^{-1}$ for airborne measurements (Lack et al., 2012a).

### 2.3.4 Additional measurements

Nephelometer measurements (TSI 3563) were used to derive the aerosol asymmetry parameter needed to apply the Müller et al. (2014) correction scheme (see Sect. 2.1.3) and were corrected according to Müller et al. (2011). A Time-Of-Flight Aerosol Mass Spectrometer (TOF-AMS) (e.g. Drewnick et al., 2005) measured the aerosol composition. The TOF-AMS was run as described in previous publications (e.g. Morgan et al., 2010).

### 2.3.5 Data Averaging

All absorption, scattering and extinction coefficient data measured using the PAS, TAP, CRDS and nephelometer were recorded at 1 Hz. Data were subsequently averaged to 30 seconds during post-flight analysis to reduce the noise in these measurements and to aid temporal alignment of the PAS and TAP for direct comparisons. To account for time lags between the PAS and TAP, an optimisation routine was run that maximised the correlation coefficient (R$^2$) between the absorption coefficients determined using the PAS and TAP by delaying one instrument relative to the other. The delay time between the TAP relative to the PAS was 20 s. There was no time lag between the PAS and CRDS when using an averaging time of 30 seconds. Time alignment was verified by visually confirming that the rising and falling edges of the peaks in the absorption coefficients aligned.

### 2.3.6 Flights and meteorology

This study uses data collected aboard the FAAM aircraft during 30 research flights (each 3-4 hours duration) in three distinct regions: London (three flights, 17 to 20 July 2017, from 1.7° W to 2.0° E and from 50.6° to 52.9° N), West Africa (three flights, 28 February to 1 March 2017, from 14.2° to 17.6 °W and from 9.6° to 14.8° N) and the Southeast Atlantic Ocean (24 flights, 16 August to 7 September 2017, from 8.0° to 18.6° W and from 4.6° N to 10.9° S). Figure 2 shows a map with the flight tracks indicated. All flights involved straight and level runs as well as deep profiles. Also shown in Fig. 2 are the mean aerosol optical depths (AODs) measured using the Moderate Resolution Imaging Spectroradiometer (MODIS) instruments aboard the Terra and Aqua satellite platforms (Remer et al., 2013) for each measurement period. The mean AOD for each region is shown corresponding to all satellite overpasses during the flight periods for both MODIS instruments. Figure 2 also shows time series of the columnar AOD values measured using the Aerosol Robotic Network (AERONET) for the Chilbolton and Oxford (~ 95 km southwest and northwest of London respectively), Dakar (West Africa) and Ascension Island (Southeast Atlantic Ocean) sites.

**Urban emissions:** during 17-20$^{th}$ July 2017, back trajectory analysis shows north-westerly flow brought air masses from over the Irish Sea to London (Rolph et al., 2017; Stein et al., 2015; available at http://ready.arl.noaa.gov/HYSPLIT_traj.





php). Flights provided measurements of regional background aerosol (Northwest London) as well as the London pollution plume (Southeast London). AOD values of ~0.00–0.13 were measured using the AERONET sites at Chilbolton and Oxford during the measurement period, as shown in Fig. 2. Mean in-flight carbon monoxide (CO) concentrations were 98 ppbv indicating the presence of fossil fuel burning, for example from transport emissions and industrial processes (e.g. Dentener et

al., 2001). These flights predominantly sampled the boundary layer with a maximum aircraft altitude of 2.2 km.

**Fresh biomass burning emissions**: flights over West Africa were dominated by freshly emitted BBA encountering similar conditions to those sampled during previous FAAM flight campaigns at the same time of year (e.g. DABEX; Haywood et al., 2008). Low-level flying through visible smoke plumes enabled measurements of fresh BBA within a few minutes of

emission. During the measurement period, MODIS measured mean AOD values ~ 0.5–0.7 over large swaths of West Africa, > 1.0 near to the coast and ~ 0.5–1.0 over the Atlantic Ocean offshore of West Africa and AERONET reported AOD values in the range ~ 0.5–0.9 over Dakar, as shown in Fig. 2. Many flights targeted measurements close to the source and were dominated by fresh BBA emissions. There was little influence of dust on our PAS, TAP and CRDS measurements because of the 1.3 µm impactor used. Mean in-flight CO concentrations were 175 ppbv although concentrations greater than 14000

ppbv were measured when flying through plumes close to the aerosol source, indicative of fresh biomass burning emissions (Dentener et al., 2001).

**Aged biomass burning emissions:** flights around Ascension Island sampled aged biomass burning aerosols transported from mainland Southern Africa in a general anticyclonic circulation (e.g. Garstang et al., 1996; Zuidema et al., 2016). East of

~ 8° W, MODIS reported mean AOD values generally between 0.1–0.5 and up to ~ 0.8 in the east of the area in which flights occurred. AERONET consistently measured AOD values between 0.1–0.5 over Ascension Island (the campaign base) during the entire four week measurement period. Mean CO concentrations were 126 ppbv, confirming that emission likely originated from a combustion source. Flights were performed in both the boundary layer and free troposphere. Based on HYSPLIT back trajectories, aerosols had generally undergone ~ 1 week of atmospheric transport since emission (Haywood

et al., in preparation).

### 3 Results and discussion

### 3.1 TAP-PAS comparisons

The primary result of this study is that the absorption coefficients determined using a TAP and PAS are linearly correlated and that the slope ($R_{abs}$) is dependent upon the aerosol source, measurement wavelength and the correction scheme applied to

the TAP measurements. Scatter plots showing the relationship between absorption coefficients measured simultaneously by the TAP and PAS for urban, fresh and aged BBA are shown in Fig. 3–5 respectively. Tight correlations between TAP and





PAS measurements were observed across all aerosol sources and for all correction schemes. A summary of $R_{abs}$ can be found in Table 2.

For the B1999 correction scheme, the range of TAP biases across all aerosol sources was 1.18–1.45. The smallest biases were consistently associated with 467 nm or 652 nm wavelength measurements and largest for 528 nm wavelength measurements. An interesting feature of this result is that the B1999 scheme led to the largest biases at 528 nm, which is the wavelength closest to that at which the scheme was derived.

For the V2010 correction scheme, the range of TAP biases across all aerosol sources was 1.08–1.38. The largest biases were consistently at 405 nm and smallest at 652 nm. Relative to the B1999 correction scheme, the V2010 scheme reduced the biases at 528 and 652 nm by 5–15 % while it increased the bias at 467 nm by 2–5 %, dependent on the aerosol source. The sensitivity of TAP biases to the wavelength dependent constants used in the V2010 scheme was investigated due to the mismatch in the TAP wavelengths and those for which the V2010 correction scheme was derived. Applying the single-wavelength V2010 correction scheme (i.e. applicable at all wavelengths) decreased TAP biases by 7–9 % at 467 nm, increased biases by 1 % at 528 nm and increased biases by 6–8 % at 652 nm.

For the M2014 correction scheme, the range of TAP biases across all aerosol sources was 0.99–1.17. The M2014 scheme reduced TAP biases relative to the B1999 and V2010 schemes by 7–40 % and 7–27 %, respectively, dependent on the aerosol source and wavelength. The most significant reductions in TAP biases were for urban aerosol emissions and had the most impact on measurements at 652 nm. As discussed in Sect. 2.1.3, the M2014 correction scheme applied here used the wavelength-dependent Virkkula (2010) parameterisation, in contrast to Müller et al. (2014), who applied the Virkkula et al. (2005) parameterisation. Although not shown, applying the Virkkula et al. (2005) parameterisation to TAP data in this study would act to decrease TAP biases by 3–4 % at 467 nm, increase biases by 1–2 % at 528 nm and by 3 % at 652 nm.

The $R_{abs}$ from Figures 3–5 provide the mean TAP absorption coefficient biases for all measurements corresponding to each measurement wavelength and aerosol source, but it is pertinent to examine the range of biases corresponding to individual 30-s average measurements. Examining the 10[th] and 90[th] percentiles of each dataset (see Table 2) revealed that 10 % of TAP measurements were biased by greater than 1.71–1.79, 1.46–1.70 and 1.39–1.42 for urban, fresh BBA and aged BBA when corrected using the B1999 scheme, respectively, dependent on wavelength. The M2014 scheme reduced the biases with 10 % of measurements biased greater than 1.27–1.41, 1.17–1.24 and 1.18–1.30 for urban, fresh BBA and aged BBA, respectively, dependent on wavelength.

The TAP biases exhibited a strong wavelength dependence. In general, the lowest biases were seen at 652 nm and the largest biases at 467 nm when the V2010 and M2014 schemes were applied to TAP measurements for all aerosol sources. The one





exception was when the M2014 scheme was applied to urban aerosol measurements, which led to the largest biases at wavelength 528 nm.

Perhaps the most important and robust observation is that the M2014 scheme consistently led to the lowest biases across all
measurement wavelengths and aerosol sources investigated. The largest biases were associated with TAP measurements corrected using the B1999 scheme at wavelengths 528 and 652 nm and when using the V2010 scheme at wavelength 467 nm for all aerosol sources.

### 3.2 Evaluating TAP biases as a function of the organic aerosol mass concentration

The biases of −1–45 % observed in this study are at the lower end of those measured by Lack et al. (2008) and Cappa et al.
(2008), who reported biases of 12 % to ~200 % dependent upon the OA concentration. To investigate this apparent discrepancy, we evaluated the TAP biases as a function of the OA mass concentration measured using an Aerodyne Aerosol Time of Flight Mass Spectrometer (TOF-AMS, Aerodyne Research Inc.) (e.g. Drewnick et al., 2005).

Figure 6 (a–c) shows how TAP biases vary with OA mass concentration for TAP measurements corrected using the B1999
correction scheme, for direct comparison with the Lack et al. (2008) study. The linear relationship between the PSAP biases and OA observed by Lack et al. (2008) is superimposed for reference. For urban emissions (Fig. 6a), TAP biases and OA mass are positively correlated and the trend is broadly consistent with that observed by Lack et al. (2008). There is however no correlation for fresh (Fig. 6b) or aged BBA (Fig. 6c).

TAP biases were also plotted as a function of the ratio of the mass concentrations of OA to light-absorbing carbon (LAC), denoted by $R_{OA/LAC}$. This was calculated using the method outlined by Lack et al. (2008) by (i) assuming all absorbing mass was black carbon, (ii) converting the mass absorption coefficient (MAC) of black carbon (BC) at 532 nm (7.75 $m^2g^{-1}$) to the PAS measurement wavelength 528 nm by using a BC AAE of 1 and the method outlined by Moosmüller et al. (2011) and (iii) dividing the PAS-measured absorption coefficient at wavelength 528 nm by the BC MAC at 528 nm. Hence the mass
concentration of LAC was calculated as $LAC = {b^{PAS}_{abs,528\,nm}}\big/{MAC^{BC}_{528\,nm}}$ such that $R_{OA/LAC} = {OA}\big/{LAC}$ (Bond and Bergstrom, 2006; Lack et al., 2008). Figure 6(d) shows that the TAP bias is positively correlated with $R_{OA/LAC}$ for urban aerosol emissions when TAP measurements were corrected using the B1999 correction. This is consistent with the Lack et al. (2008) observation although our study shows lower biases. A likely contributor to this difference is that, for consistency with the Lack et al. (2008) study, this analysis assumed all absorption was due to BC. In reality this is a poor assumption for BBA
emissions (e.g. Andreae and Gelencsér, 2006) and provides a maximum bound on the MAC value, a minimum bound on absorption attributed to LAC and therefore a maximum bound on $R_{OA/LAC}$. A more realistic approach would be to use the MAC value corresponding to BC plus BrC. Using a lower MAC to account for absorption contributions from both BC and





BrC would lead to smaller $R_{OA/LAC}$ values than those shown in Fig. 6 (d–f) and better agreement with the Lack et al. (2008) study. Correcting the TAP data using the M2014 correction scheme reduces the positive correlation between TAP biases and both $R_{OA}$ and $R_{OA/LAC}$ as shown in Fig. 6 (g–i). This further demonstrates the improvement provided by using the M2014 scheme.

This analysis was repeated at wavelengths of 467 nm and 652 nm. For measurements at 652 nm, where BrC absorbs relatively weakly (e.g. Andreae and Gelencsér, 2006), stronger correlations between TAP biases and $R_{OA}$ and $R_{OA/LAC}$ were seen compared to 528 nm measurements. This improved the agreement with Lack et al. (2008). For measurements at 467 nm, where BrC absorbs relatively strongly, weaker correlations between TAP biases and $R_{OA/LAC}$ were seen compared to 528

nm measurements. This reduced the agreement with Lack et al. (2008) for reasons described above. As for observations at 528 nm, TAP biases showed little dependence on $R_{OA}$ and $R_{OA/LAC}$ when corrected using the M2014 scheme at 652 nm and 467 nm.

A key result of this analysis is to show that biases observed in filter-based aerosol absorption measurements are strongly

dependent on the type of aerosol being sampled. Correlating biases to aerosol composition information may provide tight constraint for a single source study, such as that observed by Lack et al. (2008) for aerosol emissions over the Gulf of Mexico, but care must be taken when applying these findings more broadly to other aerosol types.

### 3.3 Impact of TAP biases on climate relevant parameters

We now assess the impact that the observed TAP biases may have on climate relevant parameters including the aerosol

single scattering albedo and absorption Ångström exponent. Figure 7 shows histograms of the SSA derived using PAS or TAP absorption data together with CRDS extinction data for the aerosol sources described in Sect. 2.3.6 and for the TAP corrections described in Sect. 2.1.1–2.1.3. The SSA is biased towards lower values when derived using TAP measurements, consistent with the results in Fig. 3–5 which typically show a ~0–45 % high bias in absorption. Campaign-mean SSA values derived using PAS and CRDS measurements for each measurement campaign are summarised in Table 3. The mean SSA

values derived using TAP and CRDS measurements matched those derived using PAS measurements most closely for fresh BBA, which were biased low by 0.00–0.03, dependent on measurement wavelength and the TAP correction scheme applied. The SSA values were most different for urban aerosols, which were biased low by 0.01–0.07, dependent on wavelength and the TAP correction scheme applied. This is consistent with the results in Table 2, which highlights that TAP biases were largest for urban aerosol measurements. The wavelength dependence of the TAP-derived SSA values depended on the

correction scheme applied. SSA values derived using the M2014 correction scheme agreed most closely with those derived using PAS measurements for all measurement wavelengths and correction schemes.





Similarly, Fig. 8 shows histograms of the AAE values derived by performing linear regressions between the logarithms of the PAS–measured absorption coefficients and the PAS measurement wavelengths (405–658 nm) (Moosmüller et al., 2011). It also shows the same information for the TAP-derived AAE values. The AAE values were calculated for the aerosol sources outlined in Sect. 2.3.6 and TAP correction schemes outlined in Sect. 2.1.1–2.1.3.

The AAE values were strongly dependent on the TAP correction scheme applied. Campaign-mean AAE values are summarised in Table 4, which highlights that the highest mean AAE values were associated with fresh BBA emissions and the lowest for aged BBA emissions. TAP-derived AAE values were in absolute error by ± 0.54. The V2010 scheme led to mean AAE values that were in closest agreement with the AAE values derived using PAS measurements for urban aerosols,

whereas the M2014 scheme provided the closest match for fresh BBA and the B1999 scheme for aged BBA. It is unclear why the different TAP correction schemes perform so differently for the different aerosol sources sampled. However, what is clear from this analysis is that there are large uncertainties in this important climate parameter when calculated from filter-based absorption measurements, and that these uncertainties are strongly source and correction scheme dependent.

## 4 Conclusions

Measurement artefacts in a commercially available filter-based absorption photometer (TAP) were evaluated as a function of wavelength and aerosol source. A range of correction schemes have been proposed in the literature to account for these artefacts and thus to maximise the accuracy of aerosol absorption coefficients determined using this technique, although biases can remain. Three correction schemes were evaluated, which all reduced the TAP mean bias to within −1 to +45 % of the PAS absorption, dependent upon aerosol source and wavelength. The largest biases were associated with urban aerosols

and the lowest for aged BBA. The M2014 correction scheme consistently led to the lowest biases across all wavelengths and aerosol sources. To our knowledge, this is the first study to demonstrate the improved performance of the M2014 correction scheme as a function of wavelength and across multiple aerosol sources for ambient aerosol sampling.

Biases in filter-based absorption measurements were strongly source dependent. On no occasion were the very large biases

of over 200 % noted in the Lack et al. (2008) study observed. However, we note that the aerosol types measured in the Lack et al. (2008) study were very different to those studied here, and therefore this result may well be consistent with the strong source dependence observed in the current study.

The positive bias in filter-based absorption measurements resulted in a low bias in determinations of single scattering

albedos of up to 0.07. The largest biases in SSA values were for urban aerosol measurements at wavelength 652 nm. The M2014 scheme consistently led to SSA values that were closest to those derived using PAS measurements across all wavelengths and aerosol sources.





Large discrepancies were seen between AAE values derived from PAS versus TAP measurements, the latter depending strongly on the correction scheme applied. The largest discrepancies in AAE values were for TAP measurements of urban aerosols corrected using the B1999 scheme, which were biased low by a mean absolute value of 0.54. Best agreement with

AAE values derived using PAS measurements was obtained when TAP measurements of (i) urban aerosol measurements were corrected using the V2010 scheme, (ii) fresh BBA measurements were corrected using the M2014 scheme and (iii) aged BBA measurements were corrected using the B1999 scheme. This highlights that the AAE is strongly source and correction scheme dependent.

The strong aerosol source dependence of biases observed in this study cautions against extrapolating results more widely to other aerosol types. Further analyses exploring biases in filter-based absorption coefficient measurements may help to address this issue. However, given the empirical nature of filter-based correction schemes and strong source and wavelength dependencies, even this is unlikely to fully bound uncertainties associated with filter-based absorption measurements to the high level of confidence that can be achieved using alternative methods, such as photoacoustic spectroscopy.

*Data availability.* For data related to this paper please contact Justin Langridge (justin.langridge@metoffice.gov.uk).

*Competing interests.* The authors declare that they have no conflict of interest.

*Acknowledgements.* This work was funded by the United Kingdom Met Office, United Kingdom Natural Environment Research Council (NERC), Norwegian Research Council, and the Royal Society of Chemistry. NERC funding was via the

CLARIFY-2017 proposals (NE/L013797/1) and (NE/L013584/1), MOYA proposal (NE/N015835/1), and a NERC/Met Office Industrial Case studentship (Ref. 640052003) for NWD. JMH and NWD also received support from ACBC and NetBC grants from the Research Council of Norway. MIC was supported by an Analytical Chemistry Trust Fund Tom West Fellowship. Airborne data was obtained using the BAe-146-301 Atmospheric Research Aircraft operated by Directflight Ltd and managed by FAAM, which is jointly supported by NERC and the Met Office. The authors acknowledge the dedicated

work of FAAM, Directflight and Avalon during the aircraft campaign. We thank the MODIS Science Team for the freely available Terra and Aqua level 2 AOD MODIS data at https://earthdata.nasa.gov. We also thank the AERONET team and in particular the PIs Brent Holben, Didier Tanre, Judith Jeffrey and Roy Grainger for providing the data used in this study.

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



| | 467 nm | 530 nm | 660 nm | Ave |
|---|---|---|---|---|
| $k_0$ | 0.377 | 0.358 | 0.352 | 0.362 |
| $k_1$ | −0.640 | −0.640 | −0.674 | −0.651 |
| $h_0$ | 1.16 | 1.17 | 1.14 | 1.159 |
| $h_1$ | −0.63 | −0.71 | −0.72 | −0.687 |
| $s$ | 0.015 | 0.017 | 0.022 | 0.018 |

Table 1: The values of the constants used in the Virkkula (2010) correction scheme (Virkkula, 2010).





| Aerosol source | Wavelength | B1999 | | | | V2010 | | | | M2014 parameterisation | | | |
|---|---|---|---|---|---|---|---|---|---|---|---|---|---|
| | | Slope | $R^2$ | $P_{10}$ | $P_{90}$ | Slope | $R^2$ | $P_{10}$ | $P_{90}$ | Slope | $R^2$ | $P_{10}$ | $P_{90}$ |
| Urban | 467 | 1.36 | 0.88 | 0.99 | 1.71 | 1.38 | 0.87 | 0.99 | 1.76 | 1.16 | 0.89 | 0.92 | 1.41 |
| | 528 | 1.45 | 0.89 | 1.11 | 1.79 | 1.37 | 0.88 | 1.03 | 1.70 | 1.17 | 0.88 | 0.94 | 1.40 |
| | 652 | 1.40 | 0.68 | 1.14 | 1.76 | 1.27 | 0.69 | 1.01 | 1.58 | 1.00 | 0.62 | 0.81 | 1.27 |
| Fresh BBA | 467 | 1.25 | 0.97 | 1.11 | 1.46 | 1.30 | 0.97 | 1.13 | 1.54 | 1.09 | 0.95 | 0.84 | 1.24 |
| | 528 | 1.30 | 0.97 | 1.17 | 1.53 | 1.23 | 0.97 | 1.08 | 1.44 | 1.08 | 0.96 | 0.84 | 1.22 |
| | 652 | 1.24 | 0.96 | 1.19 | 1.70 | 1.09 | 0.97 | 0.92 | 1.32 | 0.99 | 0.95 | 0.76 | 1.17 |
| Aged BBA | 467 | 1.18 | 0.99 | 1.10 | 1.39 | 1.21 | 0.99 | 1.11 | 1.42 | 1.11 | 0.98 | 0.99 | 1.30 |
| | 528 | 1.21 | 0.99 | 1.12 | 1.42 | 1.16 | 0.99 | 1.05 | 1.35 | 1.07 | 0.98 | 0.95 | 1.26 |
| | 652 | 1.18 | 0.99 | 1.11 | 1.42 | 1.08 | 0.99 | 1.00 | 1.29 | 1.01 | 0.99 | 0.89 | 1.18 |

**Table 2: A summary of the slopes ($R_{abs}$) between PAS and TAP absorption coefficients. Correlation coefficients ($R^2$) are also provided. $P_{10}$ and $P_{90}$ are the 10$^{th}$ and 90$^{th}$ percentiles of each dataset. All absorption coefficients correspond to > 1 Mm$^{-1}$. All linear regressions were forced through the origin.**





| Aerosol source | Wavelength | Mean SSA | | | |
|---|---|---|---|---|---|
| | | PAS | B1999 | V2010 | M2014 |
| Urban | 467 | 0.89 | 0.86 | 0.86 | 0.88 |
| | 528 | 0.88 | 0.84 | 0.85 | 0.87 |
| | 652 | 0.88 | 0.81 | 0.83 | 0.87 |
| Fresh BBA | 467 | 0.92 | 0.89 | 0.89 | 0.91 |
| | 528 | 0.93 | 0.90 | 0.91 | 0.92 |
| | 652 | 0.93 | 0.91 | 0.93 | 0.93 |
| Aged BBA | 467 | 0.84 | 0.80 | 0.79 | 0.81 |
| | 528 | 0.83 | 0.79 | 0.80 | 0.81 |
| | 652 | 0.81 | 0.77 | 0.79 | 0.81 |

**Table 3: Campaign-mean single scattering albedo (SSA) derived using PAS and CRDS measurements and TAP and CRDS measurements.**



| Aerosol source | Mean AAE | | | |
|---|---|---|---|---|
| | PAS | B1999 | V2010 | M2014 |
| Urban | 1.51 | 0.97 | 1.35 | 1.75 |
| Fresh BBA | 1.91 | 1.50 | 2.27 | 2.05 |
| Aged BBA | 1.06 | 0.99 | 1.32 | 1.36 |

**Table 4: Campaign-mean absorption Ångström exponent (AAE) derived using PAS and TAP measurements.**





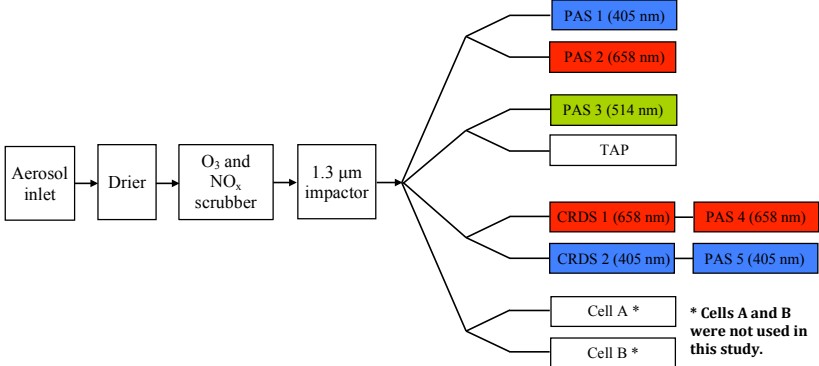

**Figure 1: Schematic diagram highlighting the flow conditioning and how the aerosol-laden stream was distributed between the PAS and CRDS cells and the TAP. All PAS and CRDS wavelengths were centred at 405, 514 and 658 nm respectively.**





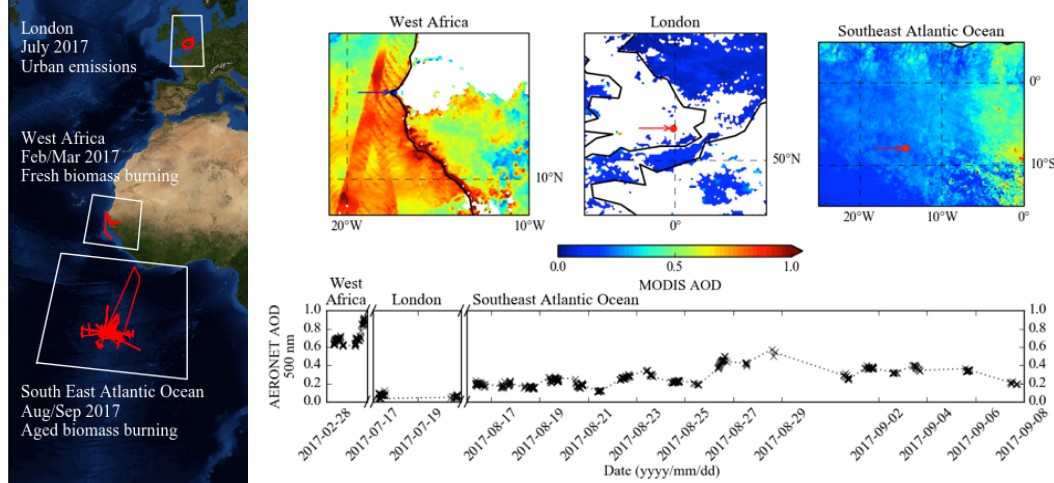

**Figure 2: FAAM research aircraft flight tracks (red) over London in the United Kingdom (July 2017), West Africa (February and March 2017) and the Southeast Atlantic (August and September 2017). For each of the geographical areas highlighted in the white boxes, the mean aerosol optical depths (AODs) measured using the Moderate Resolution Imaging Spectroradiometer (MODIS) satellite instruments are displayed. A time series of Aerosol Robotic Network (AERONET) data shows AODs at 500 nm corresponding to each measurement period. Note the discontinuous AERONET AOD time axis. AERONET sites are shown on the MODIS AOD plots by arrows.**





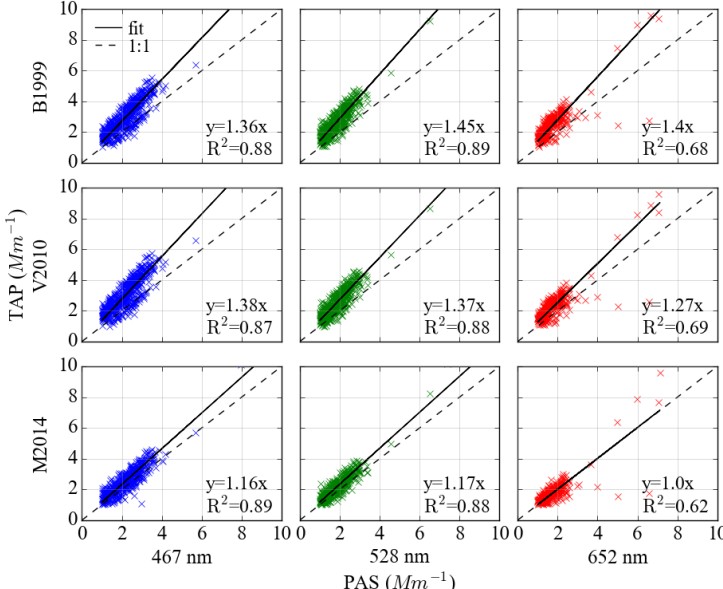

**Figure 3: Absorption coefficients measured by PAS versus TAP for urban emissions around London in July 2017. The columns correspond to: column 1: 467, column 2: 528 nm, and column 3: 652 nm wavelengths and the rows correspond to the B1999, V2010 and M2014 corrections. All absorption coefficients correspond to > 1 Mm$^{-1}$. All linear regressions were forced through the origin.**





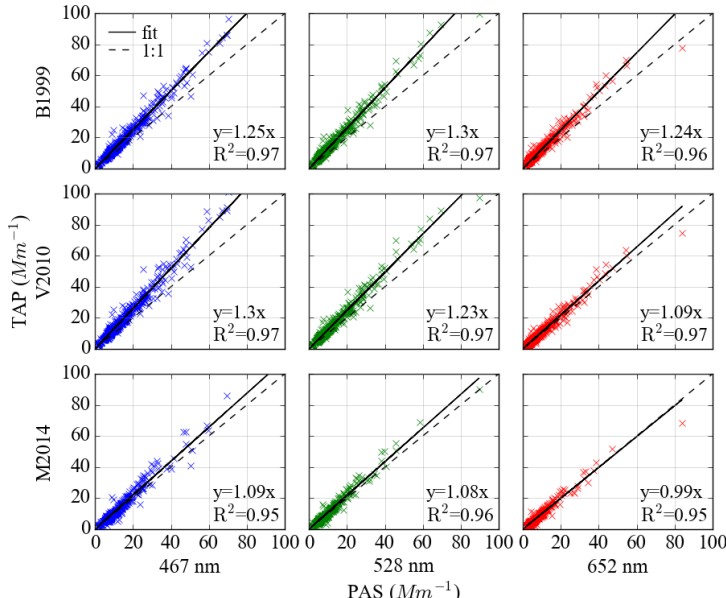

Figure 4: As Fig. 3 but for fresh biomass burning aerosol over Senegal in February and March 2017.





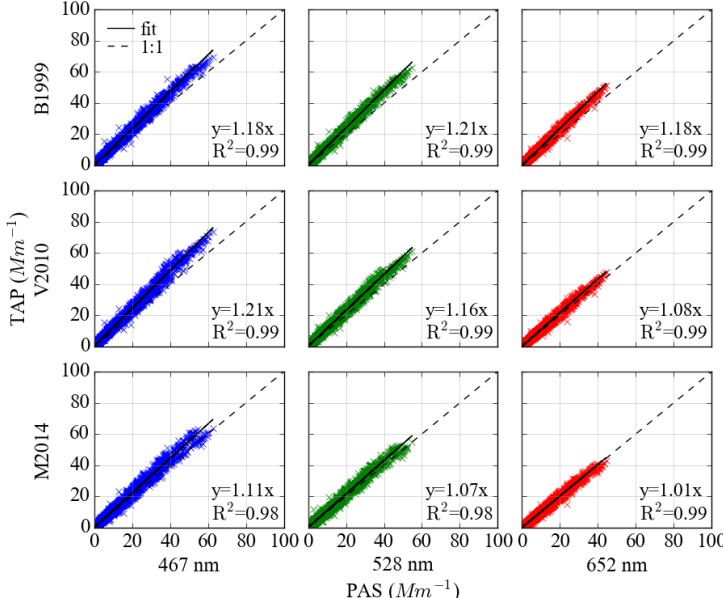

Figure 5: As Fig. 3 but for aged biomass burning aerosol over the Southeast Atlantic Ocean in August and September 2017.





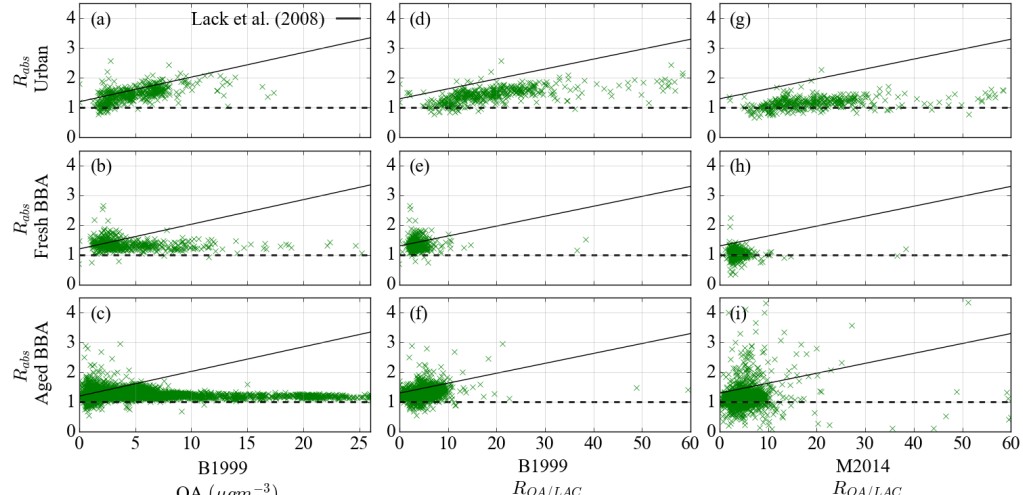

**Figure 6: The ratio of TAP to PAS absorption coefficients at 528 nm as a function of the organic aerosol mass concentration using the B1999 correction scheme (a-c) and as a function of the ratio of the organic aerosol to light-absorbing carbon mass concentrations when using the B1999 correction scheme (d-f) and using the M2014 correction scheme (g-i). All absorption coefficients correspond to > 1 Mm$^{-1}$.**



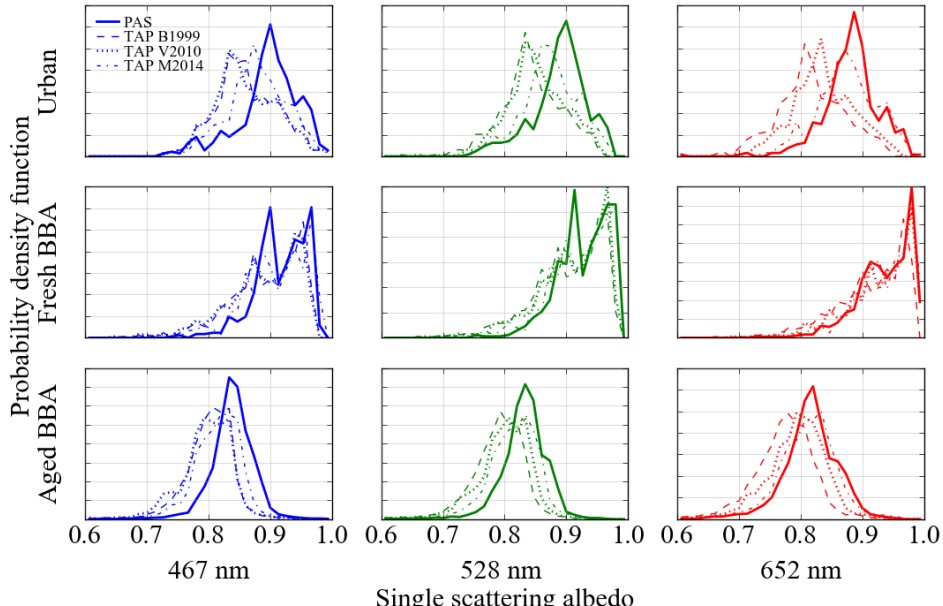

**Figure 7:** Probability density functions of the single scattering albedo derived using (i) PAS and CRDS and (ii) TAP and CRDS for the range of TAP correction schemes outlined in Sect. 2.1.1–2.1.3 at wavelengths 467, 528 and 652 nm. All absorption coefficients correspond to > 1 Mm$^{-1}$.





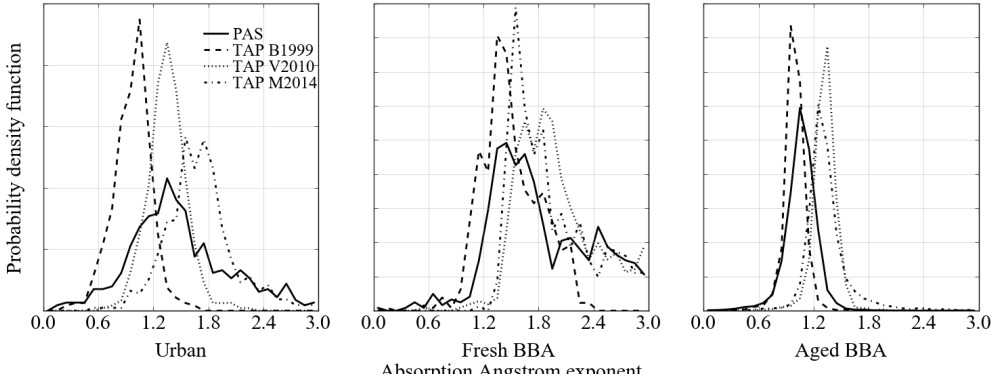

**Figure 8: Probability density functions of the absorption Ångström exponents derived for PAS and TAP measurements using the range of TAP correction schemes as outlined in Sect 2.1.1–2.1.3. All absorption coefficients correspond to > 1 Mm$^{-1}$.**