# Peer review of "Evaluating biases in filter-based aerosol absorption measurements using photoacoustic spectroscopy"

_Atmospheric Measurement Techniques, 2018_

## Referee Comment (RC1) · John Ogren (Referee) · 5 Feb 2019

This is a very good paper that should be published in AMT following the authors' consideration of the specific comments below. The logic and methods used are sound, the appropriate literature is cited, the figures and tables are clear, the length and degree of detail are appropriate, and the results are original and worthy of publication.

p.1, line 24: It's more useful here to give typical values, not extremes

p.2, line 27: Good to cite the earlier work, but Lin et al did not measure transmittance continuously during sampling, only before and after sampling.

[Figure]

p.3, line 28: recommended notation is "refractory BC" (rBC, Petzold et al., 2013)

p.5, line 3: Indeed, this is the first such study.

p.5, line 13: Recommend calling this the attenuation coefficient.

p.8, line 16: This is indeed a correct way to do the calculation, and actually was written this way in the AMTD version of the Müller et al (2014) paper (https://www.atmos-meas-tech-discuss.net/6/11093/2013/amtd-6-11093-2013.pdf). The equation 5 in Müller et al (2014) is correct, as long as one realizes that the summation is over all the particles that have been deposited on the filter, but the equation used by Davies et al. represents a practical way to do the summation.

p.8, line 23: Very confusing to use this nomenclature for the backscatter fraction, as just six lines above you use it to denote the scattering coefficient. Also, a newer version of equation 24, which is also more broadly applicable, is given by Moosmüller and Ogren (2017; Atmosphere 2017, 8, 133; doi:10.3390/atmos8080133)

p.9, line 30: Was the filtered air noise test done in-flight, on the aircraft in the hangar, or in the laboratory? If either of the latter two, are these results representative of the noise level in-flight?

p.10, line 4: This is unsubstantiated speculation. The difference could just as likely be differences between the TAP and CLAP. Since the authors have the ability to repeat the noise test with and without the digital filter, they can readily determine the effect of the digital filter on 30-s averages. Another factor suggesting a difference between the TAP and CLAP is that the present study found a pronounced difference (nearly factor of two) in the noise level of the three wavelengths, whereas Ogren et al (2017) reported that the results for the three wavelengths were very similar.

p.10, line 8: Is there a manufacturer/vendor to go along with this product number?

p.10, line 11: How does this area compare to the spot size recommended by the manufacturer? How do the less-defined spots in ambient samples affect the uncertainty of

the ambient TAP measurements?

p.10, line 30: Only four? Is there a path length correction to deal with a purge flow to prevent contamination of the PAS cell optical windows?

p.11, line 21: Are measurements on filtered air in the lab comparable to measurements on filtered air in-flight? Are there additional contributions to instrument noise from the aircraft environment, such as engine noise, vibration, turbulence, electrical interference, etc?

p.11, line 29: Wouldn't it be simpler, and more useful, to express this percentage uncertainty in terms of an absorption coefficient?

p.12, line 14: This seems like a large difference in time response. How much of the difference can be attributed to tubing lengths? Was the response difference a pure lag (i.e., corresponding to plug flow), or was there also a difference in rise/fall times that could be indicative of differences in response times due to mixing?

p.13, line 16: Not sure why you need a citation here, you could just look out the window and confirm that you were dealing with a near-source smoke plume.

p.14, line 4: Interpreting a regression slope as a bias requires that the regression intercepts are very close to zero. Please justify this implicit assumption.

p.14, lines 20-23: These comparisons provide very helpful guidance for users of M2014 in deciding which parameterisation to use for black particles. Please include the corresponding comparison of biases if you use the B1999 parameterisation for black particles in the M2014 scheme. Also, on p.6, you reported that the difference between the two flavors of V2010 (wavelength-dependent vs. independent) was minimal, but here it appears that there is a substantial difference between parameterisations for black particles. Please elaborate.

p.14, lines 27-31: How does the bias depend on filter transmittance? Is there any relationship between these 10% of points with larger biases and the filter transmittance?

Ditto for the 10% of measurements with low biases.

p.16, lines 11-12: This finding suggests that a possible contribution of BrC is not the source of the discrepancy with Lack's results, but rather that the source of the discrepancy is the correction scheme.

p.17, line 12: Please justify the claimed importance to climate of AAE. For example, what climate model, or radiative transfer model, uses AAE in the calculation of radiative forcing? I would argue that the parameter of more importance to climate is the wavelength-dependent SSA, and the results in Table 3 show that the difference between PAS and TAP+M2014 measurements of this parameter is negligible for all wavelengths and aerosol types studied. AAE is useful for inferring aerosol type, and relative contributions of BrC or dust to absorption, and it is here that the differences among the measurement approaches become important.

p.18, line 16: This should also apply to the code used to implement the various corrections, especially M2014. A brief mention of that code (i.e., what programming language) would be appropriate here. Given the conclusions of this paper, it seems likely that other users of the TAP would welcome publication of the source code for your implementation of the M2014 correction (perhaps as supplemental information?).

p.33: Interesting to see the outliers that are far below the regression line only appear in the red channel. Why don't they show up in the green channel?

---

## Referee Comment (RC2) · Anonymous Referee #2 · 5 Mar 2019

The manuscript presents an evaluation of three correction methods for filter-based particle absorption photometers using a reference method for particle light absorption. The evaluation is based on data of three different types of ambient aerosol. It is the first study of this kind for these correction methods. The study is within the scope of AMT.

The need for research on this issue has been clearly identified. The applied methods and conclusions are conclusive and clearly presented, the relevant literature has been cited. The paper is well structured and easy to read. Tables and figures reflect the key messages of the text.

The reviewer recommends the manuscript for publication in AMT after the authors have

addressed following comments.

Page 2 line 29: Particles are mainly collected in fibre filters. The penetration depth also depends on the particle size and influences the sensitivity of the photometer(Nakayama et al., 2010; Moteki et al. 2010). This circumstance should be taken into account in the discussion of the results, since different aerosols were present during the three measuring phases.

Page 9 line 11: The scrubber to remove gases may not be known to all readers. Can the author explain the function, also with the background that potentially present volatile organic material could be removed from the particles.

Page 13 line 13: The authors speculate that dust does not influence the measurements due to the impactor. Could this thesis be supported by other measurements? The reviewer assumes that the cutting characteristics refers to 1.3 $\mu$m aerodynamic diameter?

Chapter 3.3 and Figure 8: The Angström exponent strongly dependents on the source and correction schema as the author points out. The reasons cannot be clarified, but the author can deduce further motivation for this manuscript.

It is noticeable that the B1999 and also the V2010 method have the tendency to suppress large absorption Angström exponents. For the "urban" case with high R_OA/LAC ratio (c.f. figure 6) it means that the determination of the organic fraction by means of the Angström exponent would show large errors. For the "Fresh BBA" case, high absorption Angström expeonenten are measured. The R_OA/LAC ratio, on the other hand, is very low. Where do the large values for the Angström exponent come from? Could it be an indicator for mineral dust? In this case, all TAPs correction methods show large values for the Angström exponents.

A deeper aerosol characterization is not the focus of this manuscript. However, the presented results provide another good reason why this manuscript is so important.

The differentiation of aerosol types by absorption Angström exponents is becoming increasingly important, but as the data show, only with great uncertainties if filter-based photometers are used.

References:

Nakayama, T., et al. (2010). "Size-dependent correction factors for absorption measurements using filter-based photometers: PSAP and COSMOS." Journal of Aerosol Science 41(4): 333-343.

Moteki, N., et al. (2010). "Radiative transfer modeling of filter-based measurements of light absorption by particles: Importance of particle size dependent penetration depth." Journal of Aerosol Science 41(4): 401-412.

Sandradewi, J., et al. (2008). "Using aerosol light absorption measurements for the quantitative determination of wood burning and traffic emission contributions to particulate matter." Environmental Science & Technology 42(9): 3316-3323.

---

## Author Comment (AC1) · 11 May 2019

**Evaluating biases in filter-based aerosol absorption measurements using photoacoustic spectroscopy**

Nicholas W. Davies, Cathryn Fox, Kate Szpek, Michael I. Cotterell, Jonathan W. Taylor, James D. Allan, Paul I. Williams, Jamie Trembath, Jim M. Haywood, and Justin M. Langridge

We would like to thank the reviewers for taking the time to read our manuscript and for highlighting some important issues, which will be addressed in turn below.

In the responses below, an italic font highlights text from the manuscript, and underlined font and strike-through font highlight additions and deletions to the manuscript's text in response to the reviewers' comments, respectively.

**Review 1 – John Ogren**

1. p.1, line 24: It's more useful here to give typical values, not extremes

We have amended original page 1, lines 23-24, which now state:

"*Filter-based absorption measurement biases led to aerosol single-scattering albedos that were biased low by* values in the range *up to* 0.00–0.07 *and absorption Ångström exponents (AAE) that were in error by ± (*0.03–0.54*)."*

2. p.2, line 27: Good to cite the earlier work, but Lin et al did not measure transmittance continuously during sampling, only before and after sampling.

We have amended original page 2, lines 25-27, which now state:

"*Filter-based instruments measure the light transmittance across a filter continuously, which changes as particles are deposited onto the filter, providing a measure of aerosol absorption (see Sect. 2.1) (*Lin et al., 1973 *e.g. Bond et al., 1999)."*

3. p.3, line 28: recommended notation is "refractory BC" (rBC, Petzold et al., 2013)

We have amended original page 3, lines 26-28, which now state:

"*More recently, Subramanian et al. (2010) derived the BC mass absorption coefficient (MAC) at 660 nm for fresh and 1–2 day-old aerosol emissions in and around Mexico City by dividing the absorption coefficients measured using a PSAP*

*by the refractory BC mass concentrations measured using a single particle soot photometer (SP2, Droplet Measurement Technologies)."*

4. p.5, line 3: Indeed, this is the first such study.

We have amended original page 5, lines 3-5, which now state:

*" This is the first study to simultaneously evaluate the Bond et al. (1999), Virkkula (2010) and Müller et al. (2014) correction schemes for ambient aerosol sampling across multiple aerosol types."*

5. p.5, line 13: Recommend calling this the attenuation coefficient.

We have amended original page 5, line 13, which now states:

*"The  attenuation coefficient can thus be determined using"*

We have also amended original page 9, line 31, which now states:

*"Uncorrected  attenuation coefficients, $b_{ap}^{raw}$, were calculated at 1 Hz, and the average and standard deviation for…"*

6. p.8, line 16: This is indeed a correct way to do the calculation, and actually was written this way in the AMTD version of the Müller et al (2014) paper (https://www.atmos-meas- tech-discuss.net/6/11093/2013/amtd-6-11093-2013.pdf). The equation 5 in Müller et al (2014) is correct, as long as one realizes that the summation is over all the particles that have been deposited on the filter, but the equation used by Davies et al. represents a practical way to do the summation.

We have amended original page 8, lines 17-19, which now state:

*" Equation 24 is  a practical way to apply  equation 5 presented  in Müller et al. (2014) who  instead  used an equivalent method, which utilised individual particle scattering cross sections (as opposed to ensemble scattering coefficients). We used Eq. 24 as opposed to the recommended formulation because nephelometer measurements represent an ensemble."*

7. p.8, line 23: Very confusing to use this nomenclature for the backscatter fraction, as just six lines above you use it to denote the scattering coefficient. Also, a newer version of equation 24, which is also more broadly applicable, is given by Moosmüller and Ogren (2017; Atmosphere 2017, 8, 133; doi:10.3390/atmos8080133)

We have replaced "$b_{sp}$" with "$b_{back-sp}$" in equation 25 (original equation 24) on original page 8, line 22 and on original page 8, line 23 to represent the backscattering coefficient. The scattering coefficient is still represented by "$b_{sp}$" throughout the manuscript.

We have also updated our analysis to use a newer version of equation 25 (original equation 24), i.e. using equation 10 presented in Moosmüller and Ogren (2017). Using the updated equation 25 (original equation 24) only changed the results relating to the Müller et al. (2014) correction by a maximum of 1 % (i.e. the slopes in Table 2). Thus using an updated parameterisation for the asymmetry parameter does not change the overall conclusions of this manuscript. We have updated the relevant numbers in Tables 2 – 4 and throughout the manuscript to reflect this alteration (these manuscript changes are listed under point 1 in "Additional corrections" below), while our conclusions remain unchanged. Line 22 now states:

$$"g_p =  6.347 b_{back-sp}^3 +  6.906 b_{back-sp}^2 -  3.859 b_{back-sp} +  0.9852,$$

where $b_{back-sp}$ is the backscattering ratio measured using a nephelometer  _(Moosmüller and Ogren, 2017)_."

8. p.9, line 30: Was the filtered air noise test done in-flight, on the aircraft in the hangar, or in the laboratory? If either of the latter two, are these results representative of the noise level in-flight?

The filtered noise test was performed in the laboratory. Original page 9, line 30 has been amended, which now states:

_"To understand the impact of this on instrument sensitivity, the TAP was run for ~3 h in the laboratory while it sampled filtered room air to characterise the noise in the system."_

It is unclear how the noise levels will vary between laboratory and in-flight filter measurements.

9. p.10, line 4: This is unsubstantiated speculation. The difference could just as likely be differences between the TAP and CLAP. Since the authors have the ability to repeat the noise test with and without the digital filter, they can readily determine the effect of the digital filter on 30-s averages. Another factor suggesting a difference between the TAP and CLAP is that the present study found a pronounced difference (nearly factor of two) in the noise level of the three wavelengths, whereas Ogren et al (2017) reported that the results for the three wavelengths were very similar.

We have amended original page 10, lines 3-5 to state:

_"The difference between the detection limits measured in this study and that presented in Ogren et al. (2017)  could be due to running without low-pass digital filtering in the current study and/or due to differences between the TAP and CLAP."_

10. p.10, line 8: Is there a manufacturer/vendor to go along with this product number?

Yes – Sigma Aldrich. We have amended page original 10, line 8, which now states:

*"nigrosin (Sigma Aldrich, product number 198285-100G)".*

11. p.10, line 11: How does this area compare to the spot size recommended by the manufacturer? How do the less-defined spots in ambient samples affect the uncertainty of the ambient TAP measurements?

We have added the following information to original page 10, lines 11-16:

*"The manufacturer-recommended spot sizes are 30.7210 mm$^2$. Filter spot sizes were determined using nigrosin rather than from the ambient aerosol samples themselves as the spot edges were more clearly defined. The spot edges of the deposited ambient aerosol were difficult to detect as the filter spot was changed at the start of each day when measurements were taken. It was possible to detect the aerosol spot for measurements that corresponded to high loadings of absorbing aerosol. In these cases there was evidence of aerosols spreading across the filter and the area of the spots was larger by 5–20 %. However, this observation is based on a limited sample of three aerosol spots and the timescale for spread across the filter is unclear. This analysis used the areas determined using the clearly defined nigrosin spots, and therefore provides a lower limit of area , absorption coefficient (see Eq. 2), and as will be shown in Sect. 3, the TAP absorption bias."*

12. p.10, line 30: Only four? Is there a path length correction to deal with a purge flow to prevent contamination of the PAS cell optical windows?

The phrase "determined" in line 30 (page 10) is intended to mean that the uncertainty in PAS measurements is governed by the three factors listed, i.e. (i) uncertainty in the ozone calibration, (ii) uncertainty in corrections applied to account for the PAS microphone pressure sensitivity and (iii) uncertainty in subtraction of background noise which arose primarily from laser heating of the PAS cell optical windows. Other corrections to the PAS data are relatively small and therefore contribute negligibly to the overall uncertainty in PAS measurements.

We have amended original page 10, line 30, which now states:

*"The accuracy of PAS absorption measurements was determined primarily by three factors: …"*

We do not use a purge flow to prevent contamination of the PAS cell windows, as contamination during normal ambient sampling is not significant. Even then, we record window-generated background noise at relevant intervals to remove any such contribution.

13. p.11, line 21: Are measurements on filtered air in the lab comparable to measurements on filtered air in-flight? Are there additional contributions

to instrument noise from the aircraft environment, such as engine noise, vibration, turbulence, electrical interference, etc?

The noise performance was no worse than a factor of 2 larger for airborne operation, which bounded the error introduced by the background correction to be 0.27 – 0.54 Mm$^{-1}$. This updated uncertainty range (0.27–0.54 Mm$^{-1}$ compared to the originally-stated 0.2 Mm$^{-1}$) reflects a more robust uncertainty analysis, which was derived using a larger range of absorption coefficient data for multiple wavelength PAS channels.

We have amended original page 11, lines 28-29, which now state:

*"The uncertainty  was found to be  0.27–0.54 Mm⁻¹, which led to larger percentage uncertainties for lower absorption coefficients. The noise performance was no worse than a factor of 2 larger for airborne operation."*

We have also amended original page 11, line 26-27, which now states:

*"The uncertainty in the background noise correction was found to be normally distributed, with a 1σ width of  1.81–2.30 % across the range of cells used."*

We have also amended original page 11, lines 33-34, which now state:

*"These factors were combined in quadrature, leading to total PAS measurement uncertainties of  29.0–55.0 % for 1 Mm⁻¹ absorption coefficient measurements across the range of cells used (independent of pressure) and approximately  8.1 % for 100 Mm⁻¹."*

14. p.11, line 29: Wouldn't it be simpler, and more useful, to express this percentage uncertainty in terms of an absorption coefficient?

We have amended original page 11, line 29, which now states:

*"The uncertainty  was found to be  0.27–0.54 Mm⁻¹, which led to larger percentage uncertainties for lower absorption coefficients."*

15. p.12, line 14: This seems like a large difference in time response. How much of the difference can be attributed to tubing lengths? Was the response difference a pure lag (i.e., corresponding to plug flow), or was there also a difference in rise/fall times that could be indicative of differences in response times due to mixing?

The originally stated 20 seconds lag time between the TAP and PAS cells was dominated by inaccurate synchronisation between the two computers used to run the TAP and PAS instruments, respectively. The rise and fall times of the TAP and PAS were comparable. We have removed reference to the lag time between the two instruments.

We have deleted the following line from original page 12, line 14:

*"The delay time between the TAP relative to the PAS was 20 s."*

16. p.13, line 16: Not sure why you need a citation here, you could just look out the window and confirm that you were dealing with a near-source smoke plume.

We have removed the citation on original page 13, line 16, which now states:

*"Mean in-flight CO concentrations were 175 ppbv although concentrations greater than 14000 ppbv were measured when flying through plumes close to the aerosol source, indicative of fresh biomass burning emissions ."*

17. p.14, line 4: Interpreting a regression slope as a bias requires that the regression intercepts are very close to zero. Please justify this implicit assumption.

We have added the following text to original page 13, line 1:

*"All linear regressions between TAP and PAS measurements were forced through the origin."*

This information is also stated in the captions of Table 2 and Figure 3.

18. p.14, lines 20-23: These comparisons provide very helpful guidance for users of M2014 in deciding which parameterisation to use for black particles. Please include the corresponding comparison of biases if you use the B1999 parameterisation for black particles in the M2014 scheme. Also, on p.6, you reported that the difference between the two flavors of V2010 (wavelength-dependent vs. independent) was minimal, but here it appears that there is a substantial difference between parameterisations for black particles. Please elaborate.

We have updated Tables 2 to 4 and Figures 3 to 5 and 7 to 8 to reflect use of the CTS-B1999 parameterisation.

We have also included the M2014 (B1999 parameterisation) into the manuscript, replacing equation 14. The following text has been added to original page 7, line 19:

*"Using the B1999 empirical correction,*

$$F_{a,B1999}^{exp} = \frac{1}{\delta_{ap}} ln\left(\frac{e^{c_2\delta_{ap}+ln(c_1+c_2)}-c_1}{c_2}\right), \hspace{2cm} (13)$$

*where $c_1 = 1.555$ and $c_2 = 1.023$, which were derived in Bond et al. (1999); see the alternative formulation of the B1999 correction in Müller et al. (2014)."*

The other equation numbers have been updated accordingly.

We have also amended the text of original page 14, lines 17–23, which now state:

*"For the M2014 (B1999 parameterisation) correction scheme, the range of TAP biases across all aerosol sources was 1.04–1.26 and for the M2014 (V2010 parameterisation), the range of TAP biases was 1.01–1.18. The M2014 (V2010 parameterisation) scheme reduced TAP biases relative to the B1999 and V2010 schemes by 7–38 % and 7–25 %, respectively, dependent on the aerosol source and wavelength. The most significant reductions in TAP biases were for urban aerosol emissions and had the most impact on measurements at 652 nm. As discussed in Sect. 2.1.3, the M2014 (V2010 parameterisation) correction scheme applied here used the wavelength-dependent Virkkula (2010) parameterisation, in contrast to Müller et al. (2014), who applied the Virkkula et al. (2005) parameterisation."*

We have also amended the text on original page 14, line 29, which now states:

*"The M2014 (V2010 parameterisation) scheme reduced the biases with 10 % of measurements biased greater than 1.27–1.41, 1.20–1.30 and 1.18–1.29 for urban, fresh BBA and aged BBA, respectively, dependent on wavelength."*

We have also amended the text on original page 14, lines 33-34 and page 15, lines 1-2, which now state:

*"The TAP biases exhibited a strong wavelength dependence. In general, the lowest biases were seen at 652 nm and the largest biases at 467 nm when the V2010 and M2014 schemes were applied to TAP measurements for all aerosol sources. The exceptions to this trend were when the M2014 scheme (V2010 parameterisation) was applied to urban aerosol measurements, which led to the largest biases at wavelength 528 nm. The M2014 scheme (B1999 parameterisation) also led to the largest biases at 528 nm for all aerosol types."*

We have amended original page 16, line 2, which now states:

*"Correcting the TAP data using the M2014 (V2010 parameterisation) correction scheme reduces the positive correlation between TAP biases and both $R_{OA}$ and $R_{OA/LAC}$ as shown in Fig. 6 (g–i)."*

We have amended original page 17, lines 8-10, which now state:

*"The M2014 (B1999 parameterisation) led to mean AAE values that were in closest agreement with AAE values derived using PAS measurements for all aerosol types. The V2010 scheme led to mean AAE values that were in second-closest agreement with the AAE values derived using PAS measurements for urban aerosols, whereas the M2014 (V2010 parameterisation) scheme provided the second-closest match for fresh BBA and the B1999 scheme for aged BBA."*

We have also amended original page 18, lines 4-7, which now state:

*"Best agreement with AAE values derived using PAS measurements was obtained when TAP measurements were corrected using the M2014 (B1999*

*parameterisation) correction scheme and when  (i) urban aerosol measurements were corrected using the V2010 scheme, (ii) fresh BBA measurements were corrected using the M2014 scheme and (iii) aged BBA measurements were corrected using the B1999 scheme."*

We agree with the reviewer's comment relating to the difference between the wavelength-dependent and wavelength-independent flavours of the CTS-V2010 parameterisation and have updated the text on page 6, lines 22-23 accordingly, which now states:

*"This was found to have a  moderate impact on the results of this study as discussed in Sect. 3".*

We have also amended the caption of figure 6 to state:

*"using the M2014 (V2010 parameterisation) correction scheme"*

19. p.14, lines 27-31: How does the bias depend on filter transmittance? Is there any relationship between these 10% of points with larger biases and the filter transmittance? Ditto for the 10% of measurements with low biases.

We evaluated the TAP bias as a function of filter transmittance on a point-by-point basis for (i) all data, (ii) highest 10% of TAP biases and (iii) lowest 10% of TAP biases, and found no correlation for any dataset. Assessing the difference in the mean filter transmittance associated with the top 10% of TAP biases compared to the lowest 10% of TAP biases revealed absolute differences in filter transmittance up to 0.12, which depended on the measurement wavelength and correction scheme applied.

We have added the following text to page 14, line 32:

*"An analysis of the dependence of TAP bias as a function of filter loading revealed no point-by-point dependence but potentially a weak signal in the large-scale mean such that the difference in absolute filter transmittance associated with the highest 10 % of TAP biases compared to the lowest 10 % of biases across all channels and wavelengths was up to 0.12. The filter transmittance changed over the course of a flight by a maximum of 0.21."*

20. p.16, lines 11-12: This finding suggests that a possible contribution of BrC is not the source of the discrepancy with Lack's results, but rather that the source of the discrepancy is the correction scheme.

Arguably, Figure 6 shows that the different aerosol sources led to a greater discrepancy with Lack's results than by applying a different correction scheme than applied in Lack (i.e. applying V2010 and M2014 instead of B1999). Ultimately, we will never be able to reconcile the differences between our results and those of Lack due to the different aerosol mixtures measured.

We have added the following text to page 16, line 12:

*"This finding suggests that the source of discrepancy between the results presented in this study and the results of Lack et al. (2008) (i.e. Fig. 6) may be caused by the less advanced correction scheme applied to the Lack et al. (2008) data. However, given the strong dependence of $R_{abs}$ on the aerosol type and source in Fig. 6, the bias dependence on organic fraction in the Lack et al (2008) data may well persist, independent of the correction scheme used, because of the different aerosol sources and source locations being studied."*

21. p.17, line 12: Please justify the claimed importance to climate of AAE. For example, what climate model, or radiative transfer model, uses AAE in the calculation of radiative forcing? I would argue that the parameter of more importance to climate is the wavelength-dependent SSA, and the results in Table 3 show that the difference between PAS and TAP+M2014 measurements of this parameter is negligible for all wavelengths and aerosol types studied. AAE is useful for inferring aerosol type, and relative contributions of BrC or dust to absorption, and it is here that the differences among the measurement approaches become important.

We have amended the text on original page 17, line 11-13, which now states:

*"However, what is clear from this analysis is that there are large uncertainties in this important  parameter when calculated from filter-based absorption measurements, and that these uncertainties are strongly source and correction scheme dependent. This cautions that significant uncertainties could be introduced if using the AAE to differentiate between types of aerosol."*

22. p.18, line 16: This should also apply to the code used to implement the various corrections, especially M2014. A brief mention of that code (i.e., what programming language) would be appropriate here. Given the conclusions of this paper, it seems likely that other users of the TAP would welcome publication of the source code for your implementation of the M2014 correction (perhaps as supplemental information?).

At present, we are unable to invest the time to get the code to a position suitable for sharing in the public domain but will consider this as something for the future. The code used to run the analysis presented in this manuscript was implemented in Python. We have added the following text to original page 5, line 19:

*"The code used to run the analysis presented in this manuscript, i.e. relating to the equations presented throughout this section, was implemented in Python."*

23. p.33: Interesting to see the outliers that are far below the regression line only appear in the red channel. Why don't they show up in the green channel?

It is unclear why the outliers only appear in the red channel in corresponding to urban aerosol emission data, shown in Figure 3. We were unable to determine the cause of these outliers and attributed this to instability in the 652 nm PAS cell.

**Review 2**

1. Page 2 line 29: Particles are mainly collected in fibre filters. The penetration depth also depends on the particle size and influences the sensitivity of the photometer(Nakayama et al., 2010; Moteki et al. 2010). This circumstance should be taken into account in the discussion of the results, since different aerosols were present during the three measuring phases.

We have added the following text to original page 2, line 34:

*"The sensitivity of filter-based absorption photometers is also affected by the penetration depth of particles within the filter matrix, which depends on particle size (Moteki et al., 2010; Nakayama et al., 2010)."*

We have also added the following text to original page 15, line 3:

*"As highlighted in the introduction, filter-based absorption photometers are sensitive to the particle penetration depth, which is dependent on particle size. Indeed, this sensitivity may have contributed in part to the variation in TAP biases observed for the three types of aerosol investigated during this study."*

2. Page 9 line 11: The scrubber to remove gases may not be known to all readers. Can the author explain the function, also with the background that potentially present volatile organic material could be removed from the particles.

The function of the scrubber, i.e. to remove gases, is described on page 9, line 12, which states:

"The aerosol-laden stream was first dried to < 20 % relative humidity (Permapure, PD100T-12MSS) and then **passed through a scrubber** (MAST Carbon) **to remove absorbing gaseous impurities such as ozone and nitrogen dioxide**."

The carbon monolith will not denude the aerosol organics effectively. We may get some absorption of gaseous organics in the monolith that will perturb the aerosol/gas phase semi-volatile partitioning. However, the residence time in the scrubber is too short to result in significant mass loss from the aerosol via this process. In measurements by others using carbon monoliths to remove semi-volatile aerosol components, the aerosol sample is heated to approximately 300 degrees Celsius within the monolith to volatilise organic components to the gas phase to effectively denude the particles.

3. Page 13 line 13: The authors speculate that dust does not influence the measurements due to the impactor. Could this thesis be supported by other measurements? The reviewer assumes that the cutting characteristics refers to 1.3 μm aerodynamic diameter?

The reviewer is correct – the diameter is indeed an aerodynamic diameter. We have updated page 13, line 13 (relating to fresh BBA), which now states:

*" The impact of dust on our PAS, TAP and CRDS measurements was minimised because of the 1.3 μm aerodynamic impactor used. Based on the scattering Ångström exponent, there was likely a dust influence on this fresh BBA dataset."*

4. Chapter 3.3 and Figure 8: The Angström exponent strongly dependents on the source and correction schema as the author points out. The reasons cannot be clarified, but the author can deduce further motivation for this manuscript.

   It is noticeable that the B1999 and also the V2010 method have the tendency to suppress large absorption Angström exponents. For the "urban" case with high R_OA/LAC ratio (c.f. figure 6) it means that the determination of the organic fraction by means of the Angström exponent would show large errors. For the "Fresh BBA" case, high absorption Angström expeonenten are measured. The R_OA/LAC ratio, on the other hand, is very low. Where do the large values for the Angström exponent come from? Could it be an indicator for mineral dust? In this case, all TAPs correction methods show large values for the Angström exponents.

   A deeper aerosol characterization is not the focus of this manuscript. However, the presented results provide another good reason why this manuscript is so important. The differentiation of aerosol types by absorption Angström exponents is becoming increasingly important, but as the data show, only with great uncertainties if filter-based photometers are used.

We have emphasised the importance of the findings of this manuscript in relation to the absorption Ångström exponent on original page 17, lines 11-13, which state:

*"However, what is clear from this analysis is that there are large uncertainties in this important parameter when calculated from filter-based absorption measurements, and that these uncertainties are strongly source and correction scheme dependent. This cautions that significant uncertainties could be introduced if using the AAE to differentiate between types of aerosol."*

Indeed, the findings of this manuscript in relation to the variability in AAE determined using filter-based absorption measurements may serve as a suitable motivation for studying this further.

We agree with the reviewer in that the large absorption Ångström exponents (AAE) tend to be suppressed for the B1999 and V2010 correction schemes for all aerosol types (except for the V2010 scheme for fresh BBA measurements). However, the AAE values presented in Figure 8 will not affect the results of Figure 6. The organic aerosol (OA) mass concentrations were determined from aerosol mass spectroscopy and the light absorbing carbon (LAC) mass concentrations were determined by converting the mass absorption coefficient (MAC) value of black carbon (BC) at 532 nm to 528 nm using a standard AAE value for BC of 1. Neither of these measurements used the AAE measurements of Figure 6 and, therefore, cannot have led to large uncertainties in $R_{OA/LAC}$.

We agree that large AAE values were measured for fresh BBA. As the reviewer has pointed out, this result could be an indication that the measurements were contaminated with mineral dust. However, large AAE values could also suggest the presence of absorbing organic aerosols. An analysis of the scattering Ångström exponent revealed a potential dust contribution to the MOYA aerosol sample. Please refer to the amendment made in response to point 3 above.

**Additional corrections**

1. In addition to the above changes, we have updated the processing script to account for a minor coding issue related to processing of scattering coefficients and subsequent application of these scattering measurements in the M2014 correction scheme. The impact of these corrections on the results is a maximum of 4 % and, therefore, the conclusions of this manuscript are unchanged. This has a minor impact on a subset of the single scattering albedos, which, similarly, does not change the results of the manuscript. There was no impact on the AAE values. The Tables and Figures have been updated accordingly. The corresponding numbers have been updated throughout the manuscript, as detailed below.

We have amended the text on original page 1, line 20, which now states:

*"… consistently reduced biases to  0–18 % at all wavelengths"*

We have amended the text on original page 14, line 11, which now states:

*"… it increased the bias at 467 nm by  3–5 %, …"*

We have amended the text on original page 14, line 17, which now states:

*"… the range of TAP biases was  1.01–1.18 …"*

We have amended the text on original page 14, line 18, which now states:

*"… relative to the B1999 and V2010 schemes by  7–38 % and  7–25 %, …"*

We have amended the text on original page 14, lines 27-31, which now states:

*"… were biased by greater than  1.67–1.80, 1.46–1.70 and 1.39–1.42 for urban, fresh BBA and aged BBA when corrected using the B1999 scheme, respectively, dependent on wavelength. The M2014 (V2010 parameterisation) scheme reduced the biases with 10 % of measurements biased greater than 1.27–1.41,  1.20–1.30 and  1.18–1.29 for urban, fresh BBA and aged BBA, respectively, dependent on wavelength."*

We have amended the text on original page 14-15, lines 33-34 and 1-2, which now states:

*"The  exceptions to this trend were when the M2014 scheme (V2010 parameterisation) was applied to urban aerosol measurements, which led to the largest biases at wavelength 528 nm. The M2014 scheme (B1999 parameterisation) also led to the largest biases at 528 nm for all aerosol types."*

We have amended the text on original page 15, line 9, which now states:

*"The biases of  1–45 % observed in this study …"*

We have amended the text on original page 16, line 23, which now states:

*"… which typically show a ~ 1–45 % high bias in absorption."*

2. To determine TAP absorption coefficients, we first averaged the light transmitted through the TAP filter spots to 30 seconds and then input these averaged intensity values into the standard equations used to generate TAP absorption coefficients (i.e. equations 1 to 9). To clarify that the TAP absorption coefficients *were not* calculated based on 1 Hz light transmission values, we have added the following to original page 12, line 11:

*"In the case of TAP measurements, the intensities of light transmitted through a filter were first averaged to 30 seconds and then input into Eq. 1–9 to determine the corresponding absorption coefficients."*

Original page 17, line 18:

*"… reduced the TAP mean bias to within  1 to +45 % of …"*

3. To account for a fault with a calibration unit, the AMS mass concentrations have been lowered by 38 %. This update only affects the results presented in Figure 6, which has the effect of moving the points to lower OA mass concentrations (i.e. moving points towards the left). This update does not alter the conclusions presented in section 3.2 or of the overall conclusions of this paper. We have updated Figure 6 accordingly.

4. Since this manuscript has been under review, a relevant paper has been published, which we have added reference to. We have added the following to page 11, lines 10-12:

*"Moreover, our recent work has demonstrated that the calibration accuracy of PAS using ozone is optimal when the gas phase composition closely resembles that of ambient air (Cotterell et al., 2019), as is the case for calibrations performed for this work."*